# Optimal Regularization Can Mitigate Double Descent

**Preetum Nakkiran**
Harvard University
`preetum@cs.harvard.edu`

**Prayaag Venkat**
Harvard University
`pvenkat@g.harvard.edu`

**Sham Kakade**
Microsoft Research & University of Washington
`sham@cs.washington.edu`

**Tengyu Ma**
Stanford University
`tengyuma@stanford.edu`

## Abstract

Recent empirical and theoretical studies have shown that many learning algorithms – from linear regression to neural networks – can have test performance that is non-monotonic in quantities such the sample size and model size. This striking phenomenon, often referred to as "double descent", has raised questions of if we need to re-think our current understanding of generalization. In this work, we study whether the double-descent phenomenon can be avoided by using optimal regularization. Theoretically, we prove that for certain linear regression models with isotropic data distribution, optimally-tuned $\ell_2$ regularization achieves monotonic test performance as we grow either the sample size or the model size. We also demonstrate empirically that optimally-tuned $\ell_2$ regularization can mitigate double descent for more general models, including neural networks. Our results suggest that it may also be informative to study the test risk scalings of various algorithms in the context of appropriately tuned regularization.

## 1 Introduction

Recent works have demonstrated a ubiquitous "double descent" phenomenon present in a range of machine learning models, including decision trees, random features, linear regression, and deep neural networks (Opper, 1995; 2001; Advani & Saxe, 2017; Spigler et al., 2018; Belkin et al., 2018; Geiger et al., 2019b; Nakkiran et al., 2020; Belkin et al., 2019; Hastie et al., 2019; Bartlett et al., 2019; Muthukumar et al., 2019; Bibas et al., 2019; Mitra, 2019; Mei & Montanari, 2019; Liang & Rakhlin, 2018; Liang et al., 2019; Xu & Hsu, 2019; Dereziński et al., 2019; Lampinen & Ganguli, 2018; Deng et al., 2019; Nakkiran, 2019). The phenomenon is that models exhibit a peak of high test risk when they are just barely able to fit the train set, that is, to *interpolate*. For example, as we increase the size of models, test risk first decreases, then increases to a peak around when effective model size is close to the training data size, and then decreases again in the overparameterized regime. Also surprising is that Nakkiran et al. (2020) observe a double descent as we increase *sample size*, i.e. for a fixed model, training the model with more data can hurt test performance.

These striking observations highlight a potential gap in our understanding of generalization and an opportunity for improved methods. Ideally, we seek to use learning algorithms which robustly improve performance as the data or model size grow and do not exhibit such unexpected non-monotonic behaviors. In other words, we aim to improve the test performance in situations which would otherwise exhibit high test risk due to double descent. Here, a natural strategy would be to use a regularizer and tune its strength on a validation set. This motivates the central question of this work:

*When does optimally tuned regularization mitigate or remove the double-descent phenomenon?*

Another motivation is the fact that double descent is largely observed for *unregularized* or *under-regularized* models in practice. As an example, Figure 1 shows a simple linear ridge regression

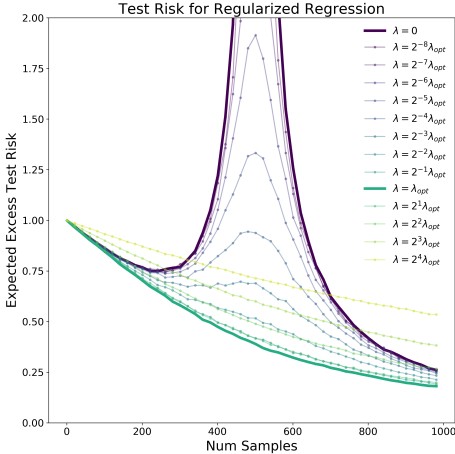

Figure 1: **Test Risk vs. Num. Samples for Isotropic Ridge Regression in** $d = 500$ **dimensions.** Unregularized regression is non-monotonic in samples, but optimally-regularized regression ($\lambda = \lambda_{opt}$) is monotonic. In this setting, the optimal regularizer $\lambda_{opt}$ does not depend on number of samples $n$ (Lemma 2), but this is not always true – see Figure 2.

setting in which the unregularized estimator exhibits double descent, but an optimally-tuned regularizer has monotonic test performance.

**Our Contributions:** We study this question from both a theoretical and empirical perspective. Theoretically, we start with the setting of high-dimensional linear regression. Linear regression is a sensible starting point to study these questions, since it already exhibits many of the qualitative features of double descent in more complex models (e.g. Belkin et al. (2019); Hastie et al. (2019) and further related works in Section 1.1). Our work shows that optimally-tuned ridge regression can achieve both sample-wise monotonicity and model-size-wise monotonicity under certain assumptions. Concretely, we show

**1. Sample-wise monotonicity:** In the setting of well-specified linear regression with isotropic features/covariates (Figure 1), we prove that optimally-tuned ridge regression yields monotonic test performance with increasing samples. That is, more data never hurts for optimally-tuned ridge regression. (See Theorem 1).

**2. Model-wise monotonicity:** We consider a setting where the input/covariate lives in a high-dimensional ambient space with isotropic covariance. Given a fixed model size $d$ (which might be much smaller than ambient dimension), we consider the family of models which first project the input to a random $d$-dimensional subspace, and then compute a linear function in this projected "feature space." (This is nearly identical to models of double-descent considered in Hastie et al. (2019, Section 5.1)). We prove that in this setting, as we grow the model-size, optimally-tuned ridge regression over the projected features has monotone test performance. That is, with optimal regularization, bigger models are always better or the same. (See Theorem 3).

**3. Monotonicity in the real-world:** We also demonstrate several richer empirical settings where optimal $\ell_2$ regularization induces monotonicity, including random feature classifiers and convolutional neural networks. This suggests that the mitigating effect of optimal regularization may hold more generally in broad machine learning contexts. (See Section 5).

A few remarks are in order:

**Problem-specific vs Minimax and Bayesian.** It is worth noting that our results hold for *all* linear ground-truths, rather than holding for only the worst-case ground-truth or a random ground-truth. Indeed, the minimax optimal estimator or the Bayes optimal estimator are both trivially sample-wise and model-wise monotonic *with respect to the minimax risk or the Bayes risk*. However, they do not guarantee monotonicity of the risk itself for a given fixed problem. In particular, there exist minimax optimal estimators which are not sample-monotonic in the sense we desire.

**Universal vs Asymptotic.** We also remark that our analysis is not only non-asymptotic but also works for all possible input dimensions, model sizes, and sample sizes. To our knowledge, the results herein are the first non-asymptotic sample-wise and model-wise monotonicity results for linear regression. (See discussion of related works Hastie et al. (2019); Mei & Montanari (2019) for related results in the asymptotic setting). Our work reveals aspects of the problem that were not

present in prior asymptotic works. For example, we empirically show that optimal regularization can eliminate even "triple descent" in ridge regression (Figure 2). Moreover, we show that for non-Gaussian covariates, optimally-tuned ridge regression is *not* always sample-monotonic: we give a counterexample in Section 4.

**Towards a more general characterization.** Our theoretical results crucially rely on the covariance of the data being isotropic. A natural next question is if and when the same results can hold more generally. A full answer to this question is beyond the scope of this paper, though we give the following results:

1. Optimally-tuned ridge regression is *not* always sample-monotonic: we show a counterexample for a certain non-Gaussian data distribution and heteroscedastic noise. We are not aware of prior work pointing out this fact. (See Section 4 for the counterexample and intuitions.)

2. For non-isotropic Gaussian covariates, we can achieve sample-wise monotonicity with a regularizer that depends on the population covariance matrix of data. This suggests unlabeled data might also help mitigate double descent in some settings, because the population covariance can be estimated from unlabeled data. (See Appendix B).

3. For non-isotropic Gaussian covariates, we conjecture that optimally-tuned ridge regression is sample-monotonic even with a standard $\ell_2$ regularizer (as in Figure 2). We derive a sufficient condition for this conjecture. Due to that current random matrix theory may be insufficient to verify this conjecture, we verify it numerically on a wide variety of cases. (See Appendix B for details).

The last two results above highlight the importance of the form of the regularizer, which leads to the open question: "How do we design good regularizers which mitigate or remove double descent?" We hope that our results can motivate future work on mitigating the double descent phenomenon, and allow us to train high performance models which do not exhibit nonmonotonic behaviors.

## 1.1 RELATED WORKS

The study of nonmonotonicity in learning algorithms existed prior to double descent and has a long history going back to (at least) Trunk (1979) and LeCun et al. (1991); Le Cun et al. (1991), where the former was largely empirical observations and the latter studied the sample non-nonmonotonicity of unregularized linear regression in terms of the eigenspectrum of the covariance matrix; the difference to our works is that we study this in the context of optimal regularization. In fact, Duin (1995; 2000); Opper (2001); Loog & Duin (2012). Loog et al. (2019) introduces the same notion of risk monotonicity which we consider, and studies several examples of monotonic and non-monotonic procedures.

Double descent of test risk as a function of model size was considered recently in more generality by Belkin et al. (2018). Similar behavior was observed empirically in earlier work in somewhat more restricted settings Trunk (1979); Opper (1995; 2001); Skurichina & Duin (2002); Le Cun et al. (1991); LeCun et al. (1991) and more recently in Advani & Saxe (2017); Geiger et al. (2019a); Spigler et al. (2018); Neal et al. (2018). Recently Nakkiran et al. (2020) demonstrated a generalized double descent phenomenon on modern deep networks, and highlighted "sample non-monotonicity" as an aspect of double descent.

A recent stream of theoretical works consider model-wise double descent in simplified settings—often via linear models for regression or classification. This also connects to works on high-dimensional regression in the statistics literature. A partial list of works in these areas include Belkin et al. (2019); Hastie et al. (2019); Bartlett et al. (2019); Muthukumar et al. (2019); Bibas et al. (2019); Mitra (2019); Mei & Montanari (2019); Liang & Rakhlin (2018); Liang et al. (2019); Xu & Hsu (2019); Dereziński et al. (2019); Lampinen & Ganguli (2018); Deng et al. (2019); Nakkiran (2019); Mahdaviyeh & Naulet (2019); Dobriban et al. (2018); Dobriban & Sheng (2019); Kobak et al. (2018). Of these, most closely related to our work are Hastie et al. (2019); Dobriban et al. (2018); Mei & Montanari (2019). Specifically, Hastie et al. (2019) considers the risk of unregularized and regularized linear regression in an asymptotic regime, where dimension $d$ and number of samples $n$ scale to infinity together, at a constant ratio $d/n$. In contrast, we show *non-asymptotic* results, and are able to consider increasing the number of samples for a fixed model, without scaling

both together. Mei & Montanari (2019) derive similar results for unregularized and regularized random features, also in an asymptotic limit. The non-asymptotic versions of the settings considered in Hastie et al. (2019) are almost identical to ours— for example, our projection model in Section 3 is nearly identical to the model in Hastie et al. (2019, Section 5.1). Finally, subsequent to our work, d'Ascoli et al. (2020) identified triple descent in an asymptotic setting.

## 2 SAMPLE MONOTONICITY IN RIDGE RIDGRESSION

In this section, we prove that optimally-regularized ridge regression has test risk that is monotonic in samples, for isotropic gaussian covariates and linear response. This confirms the behavior empirically observed in Figure 1. We also show that this monotonicity is not "fragile", and using larger than larger regularization is still sample-monotonic (consistent with Figure 1).

Formally, we consider the following linear regression problem in $d$ dimensions. The input/covariate $x \in \mathbb{R}^d$ is generated from $\mathcal{N}(0, I_d)$, and the output/response is generated by $y = \langle x, \beta^* \rangle + \varepsilon$ with $\varepsilon \sim \mathcal{N}(0, \sigma^2)$ for some unknown parameter $\beta^* \in \mathbb{R}^d$. We denote the joint distribution of $(x, y)$ by $\mathcal{D}$. We are given $n$ training examples $\{(x_i, y_i)\}_{i=1}^n$ i.i.d sampled from $\mathcal{D}$. We aim to learn a linear model $f_\beta(x) = \langle x, \beta \rangle$ with small population risk $R(\beta) := \mathbb{E}_{(x,y)\sim\mathcal{D}}[(\langle x, \beta \rangle - y)^2]$. For simplicity, let $X \in \mathbb{R}^{n \times d}$ be the data matrix that contains $x_i^\top$'s as rows and let $\vec{y} \in \mathbb{R}^n$ be column vector that contains the responses $y_i$'s as entries. For any estimator $\hat{\beta}_n(X, \vec{y})$ as a function of $n$ samples, define the expected risk of the estimator as:

$$\overline{R}(\hat{\beta}_n) := \mathop{\mathbb{E}}_{X,y\sim\mathcal{D}^n}[R(\hat{\beta}_n(X, \vec{y}))] \tag{1}$$

We consider the regularized least-squares estimator, also known as the ridge regression estimator. For a given $\lambda > 0$, define

$$\hat{\beta}_{n,\lambda} := \operatorname*{argmin}_{\beta} ||X\beta - \vec{y}||_2^2 + \lambda||\beta||_2^2 = (X^T X + \lambda I_d)^{-1} X^T \vec{y} \tag{2}$$

Here $I_d$ denotes the $d$ dimensional identity matrix. Let $\lambda_n^{\mathrm{opt}}$ be the optimal ridge parameter (that achieves the minimum expected risk) given $n$ samples: $\lambda_n^{\mathrm{opt}} := \operatorname{argmin}_{\lambda:\lambda\geq0} \overline{R}(\hat{\beta}_{n,\lambda})$. Let $\hat{\beta}_n^{\mathrm{opt}}$ be the estimator that corresponds to the $\lambda_n^{\mathrm{opt}}$. That is, $\hat{\beta}_n^{\mathrm{opt}} := \operatorname{argmin}_\beta ||X\beta - \vec{y}||_2^2 + \lambda_n^{\mathrm{opt}}||\beta||_2^2$. Our main theorem in this section shows that the expected risk of $\hat{\beta}_n^{\mathrm{opt}}$ monotonically decreases as $n$ increases.

**Theorem 1.** *In the setting above, the expected test risk of optimally-regularized well-specified isotropic linear regression is monotonic in samples. That is, for all $\beta^* \in \mathbb{R}^d$ and all $d \in \mathbb{N}, n \in \mathbb{N}, \sigma > 0$,*

$$\overline{R}(\hat{\beta}_{n+1}^{\mathrm{opt}}) \leq \overline{R}(\hat{\beta}_n^{\mathrm{opt}})$$

The above theorem shows a strong form of monotonicity, since it holds for every fixed ground-truth $\beta^*$, and does not require averaging over any prior on ground-truths. Moreover, it holds *non-asymptotically*, for every fixed $n, d \in \mathbb{N}$. Obtaining such non-asymptotic results is nontrivial, since we cannot rely on concentration properties of the involved random variables.

In particular, evaluating $\overline{R}(\hat{\beta}_n^{\mathrm{opt}})$ as a function of the problem parameters ($n, \sigma, \beta^*$, and $d$) is technically challenging. In fact, we suspect that a simple closed form expression does not exist. The key idea towards proving the theorem is to derive a "partial evaluation" — the following lemmas shows that we can write $\overline{R}(\hat{\beta}_n^{\mathrm{opt}})$ in the form of $\mathbb{E}[g(\gamma, \sigma, n, d, \beta^*)]$ where $\gamma \in \mathbb{R}^d$ contains the singular values of $X$. We will then couple the randomness of data matrices obtained by adding a single sample, and use singular value interlacing to compare their singular values.

**Lemma 1.** *In the setting of Theorem 1, let $\gamma = (\gamma_1, \ldots, \gamma_d)$ be the singular values of the data matrix $X \in \mathbb{R}^{n \times d}$. (If $n < d$, we pad the $\gamma_i = 0$ for $i > n$.) Let $\Gamma_n$ be the distribution of $\gamma$. Then, the expected test risk is*

$$\overline{R}(\hat{\beta}_{n,\lambda}) = \mathop{\mathbb{E}}_{(\gamma_1,\ldots\gamma_d)\sim\Gamma_n}\left[\sum_{i=1}^d \frac{||\beta^*||_2^2\lambda^2/d + \sigma^2\gamma_i^2}{(\gamma_i^2 + \lambda)^2}\right] + \sigma^2$$

From Lemma 1, the below lemma follows directly by taking derivatives to find the optimal $\lambda$.

**Lemma 2.** *In the setting of Theorem 1, the optimal ridge parameter is constant for all $n$: $\lambda_n^{\text{opt}} = \frac{d\sigma^2}{||\beta^*||_2^2}$. Moreover, the optimal expected test risk can be written as*

$$\overline{R}(\hat{\beta}_n^{\text{opt}}) = \mathop{\mathbb{E}}_{(\gamma_1,\ldots\gamma_d)\sim\Gamma_n} \left[ \sum_{i=1}^d \frac{\sigma^2}{\gamma_i^2 + d\sigma^2/||\beta^*||_2^2} \right] + \sigma^2 \tag{3}$$

Proofs of Lemma 1 and 2 are deferred to the Appendix, Section A.1. Now we are ready to prove Theorem 1.

*Proof of Theorem 1.* Let $\widetilde{X} \in \mathbb{R}^{(n+1)\times d}$ and $X \in \mathbb{R}^{n\times d}$ be any two matrices which differ by only the last row of $\widetilde{X}$. By the Cauchy interlacing theorem Theorem 4.3.4 of Horn et al. (1990) (c.f.,Lemma 3.4 of Marcus et al. (2014)), the singular values of $X$ and $\widetilde{X}$ are interlaced: $\forall i : \gamma_{i-1}(X) \geq \gamma_i(\widetilde{X}) \geq \gamma_i(X)$ where $\gamma_i(\cdot)$ is the $i$-th singular value.

If we couple $\widetilde{X}$ and $X$, it will induce a coupling $\Pi$ between the distributions $\Gamma_{n+1}$ and $\Gamma_n$, of the singular values of the data matrix for $n+1$ and $n$ samples. This coupling satisfies that $\widetilde{\gamma}_i \geq \gamma_i$ with probability 1 for $(\{\widetilde{\gamma}_i\}, \{\gamma_i\}) \sim \Pi$. Now, expand the test risk using Lemma 2, and observe that each term in the sum of Equation (4) below is monotone decreasing with $\gamma_i$. Thus:

$$\overline{R}(\hat{\beta}_n^{\text{opt}}) = \mathop{\mathbb{E}}_{(\gamma_1,\ldots\gamma_d)\sim\Gamma_n} \left[ \sum_{i=1}^d \frac{\sigma^2}{\gamma_i^2 + d\sigma^2/||\beta^*||_2^2} \right] + \sigma^2 \tag{4}$$

$$\geq \mathop{\mathbb{E}}_{(\widetilde{\gamma}_1,\ldots\widetilde{\gamma}_d)\sim\Gamma_{n+1}} \left[ \sum_{i=1}^d \frac{\sigma^2}{\widetilde{\gamma}_i^2 + d\sigma^2/||\beta^*||_2^2} \right] + \sigma^2 \tag{5}$$

$$= \overline{R}(\hat{\beta}_{n+1}^{\text{opt}}) \tag{6}$$

$\square$

By similar techniques, we can also prove that *overregularization* —that is, using ridge parameters $\lambda$ larger than the optimal value— is still monotonic. This proves the behavior empirically observed in Figure 1.

**Theorem 2.** *In the same setting as Theorem 1, over-regularized regression is also monotonic in samples. That is, for all $d \in \mathbb{N}, n \in \mathbb{N}, \sigma > 0, \beta^* \in \mathbb{R}^d$, the following holds*

$$\forall \lambda \geq \lambda^* : \quad \overline{R}(\hat{\beta}_{n+1,\lambda}) \leq \overline{R}(\hat{\beta}_{n,\lambda})$$

*where $\lambda^* = \frac{d\sigma^2}{||\beta^*||_2^2}$.*

*Proof.* In Section A.1. $\square$

## 3 MODEL-WISE MONOTONICITY IN RIDGE REGRESSION

In this section, we show that for a certain family of linear models, optimal regularization prevents model-wise double descent. That is, for a fixed number of samples, larger models are not worse than smaller models.

We consider the following learning problem. Informally, covariates live in a $p$-dimensional ambient space, and we consider models which first linearly project down to a random $d$-dimensional subspace, then perform ridge regression in that subspace for some $d \leq p$. Formally, the covariate $x \in \mathbb{R}^p$ is generated from $\mathcal{N}(0, I_p)$, and the response is generated by $y = \langle x, \theta \rangle + \varepsilon$ with $\varepsilon \sim \mathcal{N}(0, \sigma^2)$ and for some unknown parameter $\theta \in \mathbb{R}^p$. Next, $n$ examples $\{(x_i, y_i)\}_{i=1}^n$ are sampled i.i.d from this distribution. For a given model size $d \leq p$, we first sample a random orthonormal matrix $P \in \mathbb{R}^{d\times p}$ which specifies our model. We then consider models which operate on $(\widetilde{x}_i, y_i) \in \mathbb{R}^d \times \mathbb{R}$, where $\widetilde{x}_i = Px_i$. We denote the joint distribution of $(\widetilde{x}, y)$ by $\mathcal{D}$. Here, we emphasize that $p$ is some large ambient dimension and $d \leq p$ is the size of the model we learn.

For a fixed $P$, we want to learn a linear model $f_{\hat{\beta}}(\tilde{x}) = \langle \tilde{x}, \hat{\beta} \rangle$ for estimating $y$, with small mean squared error on distribution: $R_P(\hat{\beta}) := \mathbb{E}_{(\tilde{x}, y) \sim \mathcal{D}}[(\langle \tilde{x}, \hat{\beta} \rangle - y)^2]$. For $n$ samples $(x_i, y_i)$, let $X \in \mathbb{R}^{n \times p}$ be the data matrix, $\widetilde{X} = XP^T \in \mathbb{R}^{n \times d}$ be the projected data matrix and $\vec{y} \in \mathbb{R}^n$ be the responses. For any estimator $\hat{\beta}(\widetilde{X}, \vec{y})$ as a function of the observed samples, define the expected risk of the estimator as:

$$\overline{R}(\hat{\beta}) := \mathbb{E}_{P} \mathbb{E}_{\widetilde{X}, \vec{y} \sim \mathcal{D}^n} [R_P(\hat{\beta}(\tilde{X}, \vec{y})] \tag{7}$$

We consider the regularized least-squares estimator. For a given $\lambda > 0$, define

$$\hat{\beta}_{d,\lambda} := \underset{\beta}{\mathrm{argmin}} \, ||\widetilde{X}\beta - \vec{y}||_2^2 + \lambda ||\beta||_2^2 = (\widetilde{X}^T \widetilde{X} + \lambda I_d)^{-1} \widetilde{X}^T \vec{y} \tag{8}$$

Let $\lambda_d^{\mathrm{opt}}$ be the optimal ridge parameter (that achieves the minimum expected risk) for a model of size $d$, with $n$ samples: $\lambda_d^{\mathrm{opt}} := \mathrm{argmin}_{\lambda \geq 0} \overline{R}(\hat{\beta}_{d,\lambda})$. Let $\hat{\beta}_d^{\mathrm{opt}}$ be the estimator that corresponds to the $\lambda_d^{\mathrm{opt}}$, that is $\hat{\beta}_d^{\mathrm{opt}} := \mathrm{argmin}_{\beta} ||\widetilde{X}\beta - \vec{y}||_2^2 + \lambda_d^{\mathrm{opt}} ||\beta||_2^2$. Now, our main theorem in this setting shows that with optimal $\ell_2$ regularization, test performance is monotonic in model size.

**Theorem 3.** *In the setting above, the expected test risk of the optimally-regularized model is monotonic in the model size $d$. That is, for all $p \in \mathbb{N}, \theta \in \mathbb{R}^p, d \leq p, n \in \mathbb{N}, \sigma > 0$, we have*

$$\overline{R}(\hat{\beta}_{d+1}^{\mathrm{opt}}) \leq \overline{R}(\hat{\beta}_d^{\mathrm{opt}})$$

The proof of Theorem 3 is in Appendix A.2, and follows closely the proof of Theorem 1.

## 4 COUNTEREXAMPLES TO MONOTONICITY

In this section, we show that optimally-regularized ridge regression is *not* always monotonic in samples. We give a numeric counterexample in $d = 2$ dimensions, with non-gaussian covariates and heteroscedastic noise. This does not contradict our main theorem in Section 2, since this distribution is not jointly Gaussian with isotropic marginals.

**Counterexample.** Here we give an example of a distribution $(x, y)$ for which the expected error of optimally-regularized ridge regression with $n = 2$ samples is worse than with $n = 1$ samples. This counterexample is most intuitive to understand when the ridge parameter $\lambda$ is allowed to depend on the specific sample instance $(X, \vec{y})$ as well as $n$[1]. We sketch the intuition for this below. Consider the following distribution on $(x, y)$ in $d = 2$ dimensions. This distribution has one "clean" coordinate and one "noisy" coordinate. The distribution is: $(x, y) = (\vec{e}_1, 1)$ with probability 1/2, and $(x, y) = (\vec{e}_2, \pm 10)$ w.p. 1/2. Where $\pm 10$ is uniformly random independent noise. This distribution is "well-specified" in that the optimal predictor is linear in $x$: $\mathbb{E}[y|x] = \langle \beta^*, x \rangle$ for $\beta^* = [1, 0]$. However, the noise is heteroscedastic.

For $n = 1$ samples, the estimator can decide whether to use small $\lambda$ or large $\lambda$ depending on if the sampled coordinate is the "clean" or "noisy" one. Specifically, for the sample $(x, y)$: If $x = \vec{e}_1$, then the optimal ridge parameter is $\lambda = 0$. If $x = \vec{e}_2$, then the optimal parameter is $\lambda = \infty$.

For $n = 2$ samples, with probability $1/2$ the two samples will hit both coordinates. In this case, the estimator must chose a single value of $\lambda$ uniformly for both coordinates. This yields to a suboptimal tradeoff, since the "noisy" coordinate demands large regularization, but this hurts estimation on the "clean" coordinate.

It turns out that a slight modification to the above also serves as a counterexample to monotonicity when the regularization parameter $\lambda$ is chosen only depending on $n$ (and not on the instance $X, y$). The distribution is: $(x, y) = (\vec{e}_1, 1)$ w.p. 0.98 and $(x, y) = (\vec{e}_2, \pm 20)$ w.p. 0.02. This distribution has the following property.

**Theorem 4.** *There exists a distribution $\mathcal{D}$ over $(x, y)$ for $x \in \mathbb{R}^2, y \in \mathbb{R}$ with the following properties. Let $\hat{\beta}_n^{\mathrm{opt}}$ be the optimally-regularized ridge regression solution for $n$ samples $(X, \vec{y})$ from $\mathcal{D}$. Then:*

---

[1]Recall, our model of optimal ridge regularization from Section 2 only allows $\lambda$ to depend on $n$ (not on $X, \vec{y}$).

1. $\mathcal{D}$ is "well-specified" in that $\mathbb{E}_{\mathcal{D}}[y|x]$ is a linear function of $x$,

2. The expected test risk increases as a function of $n$, between $n = 1$ and $n = 2$. Specifically

$$\overline{R}(\hat{\beta}_{n=1}^{\text{opt}}) < \overline{R}(\hat{\beta}_{n=2}^{\text{opt}})$$

*Proof.* For $n = 1$ samples, it can be confirmed analytically that the expected risk $\overline{R}(\hat{\beta}_{n=1}^{\text{opt}}) < 8.157$. This is achieved with $\lambda = 400/2401 \approx 0.166597$. For $n = 2$ samples, it can be confirmed numerically (via Mathematica) that the expected risk $\overline{R}(\hat{\beta}_{n=2}^{\text{opt}}) > 8.179$. This is achieved with $\lambda = 0.642525$. $\square$

## 5 EXPERIMENTS

We now experimentally demonstrate that optimal $\ell_2$ regularization can mitigate double descent, in more general settings than Theorems 1 and 3.

### 5.1 SAMPLE MONOTONICITY

Here we show various settings where optimal $\ell_2$ regularization empirically induces sample-monotonic performance.

**Nonisotropic Regression.** We first consider the setting of Theorem 1, but with non-isotropic covariates $x$. That is, we perform ridge regression on samples $(x, y)$, where the covariate $x \in \mathbb{R}^d$ is generated from $\mathcal{N}(0, \Sigma)$ for $\Sigma \neq I_d$. As before, the response is generated by $y = \langle x, \beta^* \rangle + \varepsilon$ with $\varepsilon \sim \mathcal{N}(0, \sigma^2)$ for some unknown parameter $\beta^* \in \mathbb{R}^d$. We consider the same ridge regression estimator, $\hat{\beta}_{n,\lambda} := \text{argmin}_\beta ||X\beta - \vec{y}||_2^2 + \lambda ||\beta||_2^2$.

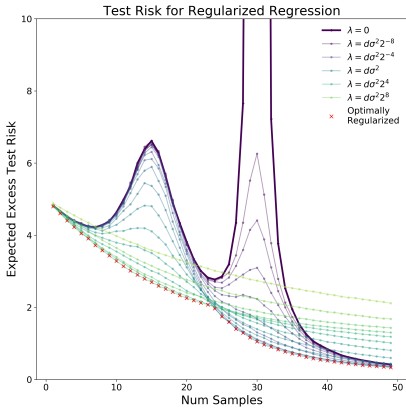

Figure 2: **Test Risk vs. Num. Samples for Non-Isotropic Ridge Regression in** $d = 30$ **dimensions.** Unregularized regression is non-monotonic in samples, but optimally-regularized regression is monotonic. Note the optimal regularization $\lambda$ depends on the number of samples $n$.

Figure 2 shows one instance of this, for a particular choice of $\Sigma$ and $\beta^*$. The covariance $\Sigma$ is diagonal, with $\Sigma_{i,i} = 10$ for $i \leq 15$ and $\Sigma_{i,i} = 1$ for $i > 15$. That is, the covariance has one "large" eigenspace and one "small" eigenspace. The ground-truth $\beta^* = 0.1\vec{e_1} + \vec{e_{30}}$, which lies almost entirely within the "small" eigenspace of $\Sigma$. The noise parameter is $\sigma = 0.5$.

We see that unregularized regression ($\lambda = 0$) actually undergoes "triple descent"[2] in this setting, with the first peak around $n = 15$ samples due to the 15-dimensional large eigenspace, and the second peak at $n = d$. In this setting, optimally-regularized ridge regression is empirically monotonic in samples (Figure 2). Unlike the isotropic setting of Section 2, the optimal ridge parameter $\lambda_n$ is no longer a constant, but varies with number of samples $n$.

---

[2]See also the "multiple descent" behavior of kernel interpolants in Liang et al. (2020).

**Random ReLU Features.** We consider random ReLU features, in the random features framework of Rahimi & Recht (2008). For a given number of features $D$, and number of samples $n$, the random feature classifier is obtained by performing regularized linear regression on the embedding $\tilde{x} := \text{ReLU}(Wx)$, where $W \in \mathbb{R}^{D \times d}$ is a matrix with each entry sampled i.i.d $\mathcal{N}(0, 1/\sqrt{d})$ and ReLU applies pointwise. This is equivalent to a 2-layer fully-connected neural network with a frozen (randomly-initialized) first layer, trained with $\ell_2$ loss and weight decay. In Appendix A.4, we apply random features to Fashion-MNIST Xiao et al. (2017). From Appendix Figure 4a, we see that underregularized models are non-monotonic, but optimal $\ell_2$ regularization is monotonic in samples. Moreover, the optimal ridge parameter $\lambda$ appears to be constant for all $n$, similar to our results from the isotropic setting in Theorem 1.

## 5.2 MODEL-SIZE MONOTONICITY

Here we empirically show that optimal $\ell_2$ regularization can mitigate model-wise double descent.

**Random ReLU Features.** We consider the same experimental setup as in Section 5.1, but now fix the number of samples $n$, and vary the number of random features $D$. This corresponds to varying the width of the corresponding 2-layer neural network. Figure 4b in Appendix A.4 shows the test error of the random features classifier, for $n = 500$ train samples and varying number of random features. We see that underregularized models undergo model-wise double descent, but optimal $\ell_2$ regularization prevents double descent.

**Convolutional Neural Networks.** We follow the experimental setup of Nakkiran et al. (2020) for model-wise double descent, and add varying amounts of $\ell_2$ regularization (weight decay). We chose the following setting from Nakkiran et al. (2020), because it exhibits double descent even with no added label noise. We consider the same family of 5-layer convolutional neural networks (CNNs) from Nakkiran et al. (2020), consisting of 4 convolutional layers of widths $[k, 2k, 4k, 8k]$ for varying $k \in \mathbb{N}$. We train and test on CIFAR-100 (Krizhevsky et al., 2009), an image classification problem with 100 classes. Inputs are normalized to

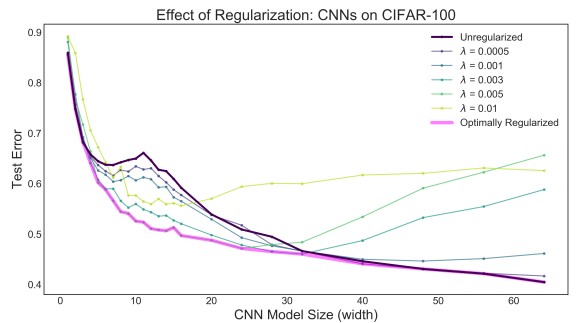

Figure 3: **Test Error vs. Model Size for 5-layer CNNs on CIFAR-100**, with $\ell_2$ regularization (weight decay). Note that the optimal regularization $\lambda$ varies with $n$.

$[-1, 1]^d$, and we use standard data-augmentation of random horizontal flip and random crop with 4-pixel padding. All models are trained using Stochastic Gradient Descent (SGD) on the cross-entropy loss, with step size $0.1/\sqrt{\lfloor T/512 \rfloor + 1}$ at step $T$. We train for $1e6$ gradient steps, and use weight decay $\lambda$ for varying $\lambda$. Due to optimization instabilities for large $\lambda$, we use the model with the minimum train loss among the last 5K gradient steps. Figure 3 shows the test error of these models on CIFAR-100. Although unregularized and under-reguarized models exhibit double descent, the test error of optimally-regularized models is largely monotonic. Note that the optimal regularization $\lambda$ varies with the model size — no single regularization value is optimal for all models.

## 6 DISCUSSION AND CONCLUSION

In this work, we study the double descent phenomenon in the context of optimal regularization. We show that, while unregularized or under-regularized models often have non-monotonic behavior, appropriate regularization can eliminate this effect.

Theoretically, we prove that for certain linear regression models with isotropic covariates, optimally-tuned $\ell_2$ regularization achieves monotonic test performance as we grow either the sample size or the model size. These are the first non-asymptotic monotonicity results we are aware of in linear

regression. We also demonstrate empirically that optimally-tuned $\ell_2$ regularization can mitigate double descent for more general models, including neural networks. We hope that our results can motivate future work on mitigating the double descent phenomenon, and allow us to train high performance models which do not exhibit unexpected nonmonotonic behaviors.

**Open Questions.** Our work suggests a number of natural open questions. First, it is open to prove (or disprove) that optimal ridge regression is sample-monotonic for non-isotropic Gaussian covariates. We conjecture that it is, and outline a potential route to proving this (via Conjectures 1 and 2 in the Appendix). Second, more broadly, it is open to prove sample-wise or model-wise monotonicity for more general (non-linear) models with appropriate regularizers. Finally, it is open to understand why large neural networks in practice are often sample-monotonic in realistic regimes of sample sizes, even without careful choice of regularization.

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

## A APPENDIX

In Section A.1 and A.2 we provide the proofs for sample-monotonicity and model-size monotonicity. In Section A.4 we include additional and omitted plots. In Section B we investigate whether monotonicity provably holds in more general models, and present a monotonicity conjecture for non-isotropic covariates.

### A.1 SAMPLE MONOTONICITY PROOFS

First we prove Lemma 1.

*Proof of Lemma 1.* For isotropic $x$, the test risk is related to the parameter error as:

$$
\begin{aligned}
R(\hat{\beta}) &:= \mathbb{E}_{(x,y)\sim\mathcal{D}}[(\langle x,\hat{\beta}\rangle - y)^2] \\
&= \mathbb{E}_{x\sim\mathcal{N}(0,I_d),\eta\sim\mathcal{N}(0,\sigma^2)}[(\langle x,\hat{\beta}-\beta^*\rangle + \eta)^2] \\
&= ||\hat{\beta}-\beta^*||_2^2 + \sigma^2
\end{aligned}
$$

Plugging in the form of $\hat{\beta}_{n,\lambda}$ and expanding:

$$
\begin{aligned}
\overline{R}(\hat{\beta}_{n,\lambda}) &:= \mathbb{E}_{X,y\sim\mathcal{D}^n}[R(\hat{\beta}_{n,\lambda})] \\
&= \mathbb{E}[||\hat{\beta}_{n,\lambda}-\beta^*||_2^2] + \sigma^2 \\
&= \mathbb{E}_{X,y}[||(X^TX+\lambda I)^{-1}X^Ty - \beta||_2^2] + \sigma^2 \\
&= \mathbb{E}_{X,\eta\sim\mathcal{N}(0,\sigma^2 I_n)}[||(X^TX+\lambda I)^{-1}X^T(X\beta^*+\eta) - \beta^*||_2^2] + \sigma^2 \\
&= \mathbb{E}_{X}[||(X^TX+\lambda I)^{-1}X^TX\beta^* - \beta^*||_2^2] + \mathbb{E}_{X,\eta}[||(X^TX+\lambda I)^{-1}X^T\eta||_2^2] + \sigma^2 \\
&= \mathbb{E}_{X}[||(X^TX+\lambda I)^{-1}X^TX\beta^* - \beta^*||_2^2] + \sigma^2\mathbb{E}_{X}[||(X^TX+\lambda I)^{-1}X^T||_F^2] + \sigma^2
\end{aligned}
$$

Now let $X = U\Sigma V^T$ be the full singular value decomposition of $X$, with $U \in \mathbb{R}^{n\times n}, \Sigma \in \mathbb{R}^{n\times d}, V \in \mathbb{R}^{d\times d}$. Let $(\gamma_1,\ldots\gamma_d)$ denote the singular values, defining $\gamma_i = 0$ for $i > \min(n,d)$. Then, continuing:

$$
\overline{R}(\hat{\beta}_{n,\lambda}) = \mathbb{E}_{V,\Sigma}[||\mathrm{diag}(\{\frac{-\lambda}{\gamma_i^2+\lambda}\})V^T\beta^*||_2^2] + \sigma^2\mathbb{E}_{\Sigma}[\sum_i \frac{\gamma_i^2}{(\gamma_i^2+\lambda)^2}] + \sigma^2 \tag{9}
$$

$$
= \mathbb{E}_{z\sim\mathrm{Unif}(||\beta^*||_2\mathbb{S}^{d-1}),\Sigma}[||\mathrm{diag}(\{\frac{-\lambda}{\gamma_i^2+\lambda}\})z||_2^2] + \sigma^2\mathbb{E}_{\Sigma}[\sum_i \frac{\gamma_i^2}{(\gamma_i^2+\lambda)^2}] + \sigma^2 \tag{10}
$$

$$
= \frac{||\beta^*||_2^2}{d}\mathbb{E}_{\Sigma}[\sum_i \frac{\lambda^2}{(\gamma_i^2+\lambda)^2}] + \sigma^2\mathbb{E}_{\Sigma}[\sum_i \frac{\gamma_i^2}{(\gamma_i^2+\lambda)^2}] + \sigma^2 \tag{11}
$$

$$
= \mathbb{E}_{\Sigma}[\sum_i \frac{||\beta^*||_2^2\lambda^2/d + \sigma^2\gamma_i^2}{(\gamma_i^2+\lambda)^2}] + \sigma^2 \tag{12}
$$

In Line (10) follows because by symmetry, the distribution of $V$ is a uniformly random orthonormal matrix, and $\Sigma$ is independent of $V$. Thus, $z := V^T\beta^*$ is distributed as a uniformly random point on the unit sphere of radius $||\beta^*||_2$.

$\square$

Next we prove Lemma 2.

*Proof of Lemma 2.* First, we determine the optimal ridge parameter. Using Lemma 1, we have

$$\frac{\partial}{\partial \lambda} \overline{R}(\hat{\beta}_{n,\lambda}) = \frac{\partial}{\partial \lambda} \mathop{\mathbb{E}}_{(\gamma_1, \dots \gamma_d) \sim \Gamma} \left[ \sum_i \frac{||\beta||_2^2 \lambda^2 / d + \sigma^2 \gamma_i^2}{(\gamma_i^2 + \lambda)^2} \right]$$

$$= 2(||\beta^*||_2^2 \lambda / d - \sigma^2) \underbrace{\mathop{\mathbb{E}}_{(\gamma_1, \dots \gamma_d) \sim \Gamma} \left[ \sum_i \frac{\gamma_i^2}{(\gamma_i^2 + \lambda)^3} \right]}_{>0}$$

Thus, $\frac{\partial}{\partial \lambda} \overline{R}(\hat{\beta}_{n,\lambda}) = 0 \implies \lambda = \frac{d\sigma^2}{||\beta^*||_2^2}$ and we conclude that $\lambda_n^{\text{opt}} = \frac{d\sigma^2}{||\beta^*||_2^2}$.

For this optimal parameter, the test risk follows from Lemma 1 as

$$\overline{R}(\hat{\beta}_n^{\text{opt}}) = \overline{R}(\hat{\beta}_{n,\lambda_n^{\text{opt}}}) \tag{13}$$

$$= \mathop{\mathbb{E}}_{(\gamma_1, \dots \gamma_d) \sim \Gamma_n} \left[ \sum_{i=1}^d \frac{\sigma^2}{\gamma_i^2 + d\sigma^2 / ||\beta^*||_2^2} \right] + \sigma^2 \tag{14}$$

$\square$

*Proof of Theorem 2.* We follow a similar proof strategy as in Theorem 1: we invoke singular value interlacing ($\widetilde{\gamma}_i \geq \gamma_i$) for the data matrix when adding a single sample. We then apply Lemma 1 to argue that the test risk varies monotonically with the singular values.

We have

$$\overline{R}(\hat{\beta}_{n,\lambda}) = \mathop{\mathbb{E}}_{(\gamma_1, \dots \gamma_d) \sim \Gamma} [\sum_i \underbrace{\frac{||\beta^*||_2^2 \lambda^2 / d + \sigma^2 \gamma_i^2}{(\gamma_i^2 + \lambda)^2}}_{S(\gamma_i)}]$$

and we compute how each term in the sum varies with $\gamma_i$:

$$\frac{\partial}{\partial \gamma_i} \sum_i S(\gamma_i) = \frac{\partial}{\partial \gamma_i} S(\gamma_i)$$

$$= (\frac{-2\gamma_i}{d}) \frac{2||\beta^*||_2^2 \lambda^2 + d\sigma^2 (\gamma_i^2 - \lambda)}{(\gamma_i^2 + \lambda)^3}$$

Thus we have

$$\lambda \geq \frac{d\sigma^2}{2||\beta^*||^2} \implies \frac{\partial}{\partial \gamma_i} S(\gamma_i) \leq 0 \tag{15}$$

By the coupling argument in Theorem 1, this implies that the test risk is monotonic:

$$\overline{R}(\hat{\beta}_{n+1,\lambda}) - \overline{R}(\hat{\beta}_{n,\lambda})$$

$$= \mathop{\mathbb{E}}_{(\widetilde{\gamma}_1, \dots \widetilde{\gamma}_d) \sim \Gamma_{n+1}} \left[ \sum_{i=1}^d S(\widetilde{\gamma}_i) \right] - \mathop{\mathbb{E}}_{(\gamma_1, \dots \gamma_d) \sim \Gamma_n} \left[ \sum_{i=1}^d S(\gamma_i) \right]$$

$$= \mathop{\mathbb{E}}_{(\{\widetilde{\gamma}_i\}, \{\gamma_i\}) \sim \Pi} \left[ \sum_{i=1}^d S(\widetilde{\gamma}_i) - S(\gamma_i) \right] \tag{16}$$

$$\leq 0 \tag{17}$$

where $\Pi$ is the coupling. Line (17) follows from Equation (15), and the fact that the coupling obeys $\widetilde{\gamma}_i \geq \gamma_i$. $\square$

## A.2 PROJECTION MODEL PROOFS

**Lemma 3.** *For all $\theta \in \mathbb{R}^p$, $d, n \in \mathbb{N}$, and $\lambda > 0$, let $X \in \mathbb{R}^{n \times p}$ be a matrix with i.i.d. $\mathcal{N}(0,1)$ entries. Let $P \in \mathbb{R}^{d \times p}$ be a random orthonormal matrix. Define $\widetilde{X} := XP^T$. Let $(\gamma_1, \ldots, \gamma_m)$ be the singular values of the data matrix $\tilde{X} \in \mathbb{R}^{n \times d}$, for $m := \max(n, d)$ (with $\gamma_i = 0$ for $i > \min(n, d)$). Let $\Gamma_d$ be the distribution of singular values $(\gamma_1, \ldots, \gamma_m)$.*

*Then, the optimal ridge parameter is constant for all $d$: $\lambda_d^{\mathrm{opt}} = \frac{p^2 \widetilde{\sigma}^2}{d\|\theta\|_2^2}$. where we define $\widetilde{\sigma}^2 := \sigma^2 + \frac{p-d}{p}\|\theta\|_2^2$. Moreover, the optimal expected test risk can be written as*

$$\overline{R}(\hat{\beta}_d^{\mathrm{opt}}) = \widetilde{\sigma}^2 + \underset{(\gamma_1,\ldots,\gamma_m)\sim\Gamma_d}{\mathbb{E}}\left[\sum_{i=1}^p \frac{\widetilde{\sigma}^2}{\gamma_i^2 + \frac{\widetilde{\sigma}^2 p^2}{d\|\theta\|_2^2}}\right]$$

*Proof.* This proof follows exactly analogously as the proof of Lemma 2 from Lemma 1, in Section A.1. $\qquad\square$

**Lemma 4.** *For all $\theta \in \mathbb{R}^p$, $d, n \in \mathbb{N}$, and $\lambda > 0$, let $X \in \mathbb{R}^{n \times p}$ be a matrix with i.i.d. $\mathcal{N}(0,1)$ entries. Let $P \in \mathbb{R}^{d \times p}$ be a random orthonormal matrix. Define $\widetilde{X} := XP^T$ and $\beta^* := P\theta$.*

*Let $(\gamma_1, \ldots, \gamma_m)$ be the singular values of the data matrix $\tilde{X} \in \mathbb{R}^{n \times d}$, for $m := \max(n, d)$ (with $\gamma_i = 0$ for $i > \min(n, d)$). Let $\Gamma_d$ be the distribution of singular values $(\gamma_1, \ldots, \gamma_m)$.*

*Then, the expected test risk is*

$$\overline{R}(\hat{\beta}_{d,\lambda}) := \underset{P}{\mathbb{E}} \, \underset{\widetilde{X},\vec{y}\sim\mathcal{D}^n}{\mathbb{E}} [R_P(\hat{\beta}_{d,\lambda}(\tilde{X}, \vec{y})]$$

$$= \sigma^2 + (1 - \frac{d}{p})\|\theta\|_2^2$$

$$+ \underset{(\gamma_1,\ldots,\gamma_m)\sim\Gamma_d}{\mathbb{E}}\left[\sum_{i=1}^p \frac{(\sigma^2 + \frac{p-d}{p}\|\theta\|_2^2)\gamma_i^2 + \frac{d}{p^2}\|\theta\|_2^2\lambda^2}{(\gamma_i^2 + \lambda)^2}\right]$$

*Proof of Lemma 4.* We first define the parameter that minimizes the population risk. It follows directly that:

$$\beta_P^* := \underset{\beta\in\mathbb{R}^d}{\arg\min} \, R_P(\beta) = P\theta$$

First, we can expand the risk as

$$R(\hat{\beta}) = \underset{(\widetilde{x},y)\sim\mathcal{D}}{\mathbb{E}}[(\langle Px, \hat{\beta}\rangle - y)^2] \tag{18}$$

$$= \underset{(\widetilde{x},y)\sim\mathcal{D},\eta\sim\mathcal{N}(0,\sigma^2)}{\mathbb{E}}[(\langle x, P^T\hat{\beta} - \theta\rangle + \eta)^2] \tag{19}$$

$$= \sigma^2 + \|\theta - P^T\hat{\beta}\|_2^2 \tag{20}$$

$$= \sigma^2 + \|\theta - P^T\beta^*\|_2^2 + \|P^T\beta^* - P^T\hat{\beta}\|_2^2 \tag{21}$$

$$+ 2\langle(\theta - P^T\beta^*), P^T\beta^* - P^T\hat{\beta}\rangle \tag{22}$$

$$= \sigma^2 + \|\theta - P^T\beta^*\|_2^2 + \|P^T\beta^* - P^T\hat{\beta}\|_2^2 \tag{23}$$

$$= \sigma^2 + \|\theta - P^TP\theta\|_2^2 + \|\beta^* - \hat{\beta}\|_2^2 \tag{24}$$

The cross terms in Line (22) vanish because the first-order optimality condition for $\beta^*$ implies that $\beta^*$ satisfies $P(\theta^* - P^T\beta^*) = 0$. We now simplify each of the two remaining terms.

First, we have that:

$$\underset{P}{\mathbb{E}}\|\theta - P^TP\theta\|_2^2 = (1 - \frac{d}{p})\|\theta\|_2^2 \tag{25}$$

since $P^T P$ is an orthogonal projection onto a random $d$-dimensional subspace.

Now, recall we have $\vec{y} = X\theta + \eta$ where $\eta \sim \mathcal{N}(0, \sigma^2 I_n)$. Expand this as:

$$\vec{y} = X\theta + \eta \tag{26}$$
$$= XP^T P\theta + X(1 - P^T P)\theta + \eta \tag{27}$$
$$= \widetilde{X}\beta^* + \varepsilon + \eta \tag{28}$$

where $\varepsilon := X(1 - P^T P)\theta$. Note that conditioned on $P$, the three terms $\widetilde{X}, \varepsilon$ and $\eta$ are conditionally independent, since $P^T P$ and $(I - P^T P)$ project $X$ onto orthogonal subspaces. And further, $\varepsilon \sim \mathcal{N}(0, ||(1 - P^T P)\theta||^2 I_n)$.

$$\mathop{\mathbb{E}}_{P} \mathop{\mathbb{E}}_{\widetilde{X},y} ||\hat{\beta} - \beta^*||_2^2 \tag{29}$$

$$= \mathop{\mathbb{E}}_{P} \mathop{\mathbb{E}}_{\widetilde{X},y} ||(\widetilde{X}^T \widetilde{X} + \lambda I)^{-1}\widetilde{X}^T y - \beta^*||_2^2 \tag{30}$$

$$= \mathop{\mathbb{E}}_{P} \mathop{\mathbb{E}}_{\widetilde{X},y,\varepsilon,\eta} ||(\widetilde{X}^T \widetilde{X} + \lambda I)^{-1}\widetilde{X}^T(\widetilde{X}\beta^* + \varepsilon + \eta) - \beta^*||_2^2 \tag{31}$$

$$= \mathop{\mathbb{E}}_{P} \mathop{\mathbb{E}}_{\widetilde{X},y,\varepsilon,\eta} [||(\widetilde{X}^T \widetilde{X} + \lambda I)^{-1}\widetilde{X}^T \widetilde{X}\beta^* - \beta^*||_2^2 \tag{32}$$

$$+ ||(\widetilde{X}^T \widetilde{X} + \lambda I)^{-1}\widetilde{X}^T \varepsilon||_2^2 + ||(\widetilde{X}^T \widetilde{X} + \lambda I)^{-1}\widetilde{X}^T \eta||_2^2] \tag{33}$$

$$\tag{34}$$

Now, since $\widetilde{X}$ is conditionally independent of $\varepsilon$ conditioned on $P$,

$$\mathop{\mathbb{E}}_{P} \mathop{\mathbb{E}}_{\widetilde{X},y,\varepsilon|P} ||(\widetilde{X}^T \widetilde{X} + \lambda I)^{-1}\widetilde{X}^T \varepsilon||_2^2 \tag{35}$$

$$= \mathop{\mathbb{E}}_{P} \mathop{\mathbb{E}}_{\widetilde{X}|P} [||(\widetilde{X}^T \widetilde{X} + \lambda I)^{-1}\widetilde{X}^T||_F^2] \mathop{\mathbb{E}}_{\varepsilon|P} [||\varepsilon||_2^2] \tag{36}$$

$$= \mathop{\mathbb{E}}_{\widetilde{X}} [||(\widetilde{X}^T \widetilde{X} + \lambda I)^{-1}\widetilde{X}^T||_F^2] \mathop{\mathbb{E}}_{P,\varepsilon} [||\varepsilon||_2^2] \tag{37}$$

$$= \mathop{\mathbb{E}}_{\widetilde{X}} [||(\widetilde{X}^T \widetilde{X} + \lambda I)^{-1}\widetilde{X}^T||_F^2] \cdot \mathop{\mathbb{E}}_{P,X} [||X(1 - P^T P)\theta||_2^2] \quad \text{(by definition of } \varepsilon\text{)}$$

$$= \mathop{\mathbb{E}}_{\widetilde{X}} [||(\widetilde{X}^T \widetilde{X} + \lambda I)^{-1}\widetilde{X}^T||_F^2] (\frac{p - d}{p} ||\theta||_2^2) \tag{38}$$

where Line (37) holds because the marginal distribution of $\widetilde{X}$ does not depend on $P$.

Similarly,

$$\mathop{\mathbb{E}}_{P} \mathop{\mathbb{E}}_{\widetilde{X},y,\eta|P} ||(\widetilde{X}^T \widetilde{X} + \lambda I)^{-1}\widetilde{X}^T \eta||_2^2 \tag{39}$$

$$= \sigma^2 \mathop{\mathbb{E}}_{\widetilde{X}} [||(\widetilde{X}^T \widetilde{X} + \lambda I)^{-1}\widetilde{X}^T||_F^2] \tag{40}$$

Now let $\widetilde{X} = U\Sigma V^T$ be the full singular value decomposition of $\widetilde{X}$, with $U \in \mathbb{R}^{n \times n}, \Sigma \in \mathbb{R}^{n \times d}, V \in \mathbb{R}^{d \times d}$. Let $(\gamma_1, \ldots \gamma_m)$ denote the singular values, where $m = \max(n, d)$ and defining $\gamma_i = 0$ for $i > \min(n, d)$.

Observe that by symmetry, $\widetilde{X} = XP^T$ and $P$ are independent, because the joint distribution $(\widetilde{X}, P)$ is equivalent to the distribution $(\widetilde{X}Q, PQ)$ for a random orthonormal $Q \in \mathbb{R}^{p \times p}$. Thus $\widetilde{X}$ and

$\beta^* = P\theta$ are also independent, and we have:

$$\mathop{\mathbb{E}}_{P}\mathop{\mathbb{E}}_{\widetilde{X}} ||(\widetilde{X}^T\widetilde{X} + \lambda I)^{-1}\widetilde{X}^T\widetilde{X}\beta^* - \beta^*||_2^2 \tag{41}$$

$$= \mathop{\mathbb{E}}_{\beta^*}\mathop{\mathbb{E}}_{V,\Sigma}[||\mathrm{diag}(\{\frac{-\lambda}{\gamma_i^2 + \lambda}\})V^T\beta^*||_2^2] \tag{42}$$

$$= \mathop{\mathbb{E}}_{\beta^*}\mathop{\mathbb{E}}_{V,\Sigma}[||\mathrm{diag}(\{\frac{-\lambda}{\gamma_i^2 + \lambda}\})V^T\beta^*||_2^2] \tag{43}$$

$$= \mathop{\mathbb{E}}_{\beta^*}\mathop{\mathbb{E}}_{z\sim\mathrm{Unif}(||\beta^*||_2\mathbb{S}^{d-1}),\Sigma}[||\mathrm{diag}(\{\frac{-\lambda}{\gamma_i^2 + \lambda}\})z||_2^2] \tag{44}$$

$$= \mathop{\mathbb{E}}_{\beta^*}[\frac{||\beta^*||_2^2}{p}\mathop{\mathbb{E}}_{\Sigma}[\sum_i \frac{\lambda^2}{(\gamma_i^2 + \lambda)^2}]] \tag{45}$$

$$= \frac{1}{p}\mathop{\mathbb{E}}_{P}[||P\theta||_2^2] \cdot \mathop{\mathbb{E}}_{\Sigma}[\sum_i \frac{\lambda^2}{(\gamma_i^2 + \lambda)^2}] \tag{46}$$

$$= \frac{d}{p^2}||\theta||_2^2\mathop{\mathbb{E}}_{\Sigma}[\sum_i \frac{\lambda^2}{(\gamma_i^2 + \lambda)^2}] \tag{47}$$

Finally, continuing from Line (33), we can use Lines (38), (40), and (47) to write:

$$\mathop{\mathbb{E}}_{P}\mathop{\mathbb{E}}_{\widetilde{X},y} ||\hat{\beta} - \beta^*||_2^2 \tag{48}$$

$$= (\sigma^2 + \frac{p-d}{p}||\theta||_2^2)\mathop{\mathbb{E}}_{\widetilde{X}}[||(\widetilde{X}^T\widetilde{X} + \lambda I)^{-1}\widetilde{X}^T||_F^2] \tag{49}$$

$$+ \frac{d}{p^2}||\theta||_2^2\mathop{\mathbb{E}}_{\Sigma}[\sum_i \frac{\lambda^2}{(\gamma_i^2 + \lambda)^2}] \tag{50}$$

$$= (\sigma^2 + \frac{p-d}{p}||\theta||_2^2)\mathop{\mathbb{E}}_{\Sigma}[\sum_i \frac{\gamma_i^2}{(\gamma_i^2 + \lambda)^2}] \tag{51}$$

$$+ \frac{d}{p^2}||\theta||_2^2\mathop{\mathbb{E}}_{\Sigma}[\sum_i \frac{\lambda^2}{(\gamma_i^2 + \lambda)^2}] \tag{52}$$

$$= \mathop{\mathbb{E}}_{\Sigma}[\sum_i \frac{(\sigma^2 + \frac{p-d}{p}||\theta||_2^2)\gamma_i^2 + \frac{d}{p^2}||\theta||_2^2\lambda^2}{(\gamma_i^2 + \lambda)^2}] \tag{53}$$

Now, we can continue from Line (24), and apply lines (25), to conclude:

$$\mathbb{E}[R(\hat{\beta})] = \sigma^2 + \mathbb{E}[||\theta - P^TP\theta||_2^2] + \mathbb{E}[||\beta^* - \hat{\beta}||_2^2]$$

$$= \sigma^2 + (1 - \frac{d}{p})||\theta||_2^2$$

$$+ \mathop{\mathbb{E}}_{\Sigma}\left[\sum_{i=1}^{p} \frac{(\sigma^2 + \frac{p-d}{p}||\theta||_2^2)\gamma_i^2 + \frac{d}{p^2}||\theta||_2^2\lambda^2}{(\gamma_i^2 + \lambda)^2}\right]$$

$$\square$$

*Proof of Theorem 3.* This follows analogously to the proof of Theorem 1, Let $\widetilde{X}_d$ and $\widetilde{X}_{d+1}$ be the observed data matrices for $d$ and $d+1$ model size. As in Theorem 1, there exists a coupling $\Pi$ between the distributions $\Gamma_d$ and $\Gamma_{d+1}$ of the singular values of $\widetilde{X}_d$ and $\widetilde{X}_{d+1}$ such that these singular values are interlaced.

Thus by Lemma 3,

$$\overline{R}(\hat{\beta}_d^{\text{opt}}) = \widetilde{\sigma}^2 + \mathop{\mathbb{E}}_{(\gamma_1,\ldots,\gamma_m)\sim\Gamma_d}\left[\sum_{i=1}^p \frac{\widetilde{\sigma}^2}{\gamma_i^2 + \frac{\widetilde{\sigma}^2 p^2}{d||\theta||_2^2}}\right]$$

$$\geq \widetilde{\sigma}^2 + \mathop{\mathbb{E}}_{(\widetilde{\gamma}_1,\ldots,\widetilde{\gamma}_m)\sim\Gamma_{d+1}}\left[\sum_{i=1}^p \frac{\widetilde{\sigma}^2}{\widetilde{\gamma}_i^2 + \frac{\widetilde{\sigma}^2 p^2}{d||\theta||_2^2}}\right]$$

$$= \overline{R}(\hat{\beta}_{d+1}^{\text{opt}})$$

$\square$

### A.3 Nonisotropic Reduction

Here we observe that results on isotropic regression in Section 2 also imply that ridge regression can be made sample-monotonic even for non-isotropic covariates, if an appropriate regularzier is applied. Specifically, the regularizer depends on the covariance on the inputs. This follows from a general equivalence between the non-isotropic and isotropic problems.

**Lemma 5.** *For all $n \in \mathbb{N}, d \in \mathbb{N}, \lambda \in \mathbb{R}, \sigma \in \mathbb{R}$, covariance $\Sigma \in \mathbb{R}^{d\times d}$, PSD matrix $M \in \mathbb{R}^{d\times d}$, and ground-truth $\beta^* \in \mathbb{R}^d$, the following holds.*

*Consider the following two problems:*

1. *Regularized regression on isotropic covariates, and an $M$-regularizer. That is, suppose $n$ samples $(x, y)$ are drawn with covariates $x \sim \mathcal{N}(0, I_d)$ and response $y = \langle \beta^*, x \rangle + \mathcal{N}(0, \sigma^2)$. Let $X \in \mathbb{R}^{n\times d}$ be the matrix of covariates, and $\vec{y}$ the vector of responses. Consider*

$$\hat{\beta}_\lambda := \operatorname*{argmin}_{\beta} ||X\beta - \vec{y}||_2^2 + \lambda ||\beta||_M^2 \tag{54}$$

*Let $\overline{R} = \mathbb{E}_{X,y}[||\hat{\beta} - \beta^*||_2^2] + \sigma^2$ be the expected test risk of the above estimator.*

2. *Regularized regression with covariance $\Sigma$, and an $(\Sigma^{1/2}M\Sigma^{1/2})$-regularizer. That is, suppose $n$ samples $(\widetilde{x}, y)$ are drawn with covariates $\widetilde{x} \sim \mathcal{N}(0, \Sigma)$ and response $y = \langle z^*, \widetilde{x} \rangle + \mathcal{N}(0, \sigma^2)$, for*

$$z^* = \Sigma^{-1/2}\beta^*$$

*Let $\widetilde{X} \in \mathbb{R}^{n\times d}$ be the matrix of covariates, and $\vec{y}$ the vector of responses. Consider*

$$\hat{z}_\lambda := \operatorname*{argmin}_{z} ||\widetilde{X}z - \vec{y}||_2^2 + \lambda ||z||_{\Sigma^{1/2}M\Sigma^{1/2}}^2 \tag{55}$$

*Let $\widetilde{R} = \mathbb{E}_{\widetilde{X},y}[||\hat{z} - z^*||_\Sigma^2] + \sigma^2$ be the expected test risk of the above estimator.*

*Then, the expected test risks of the above two problems are identical:*

$$\overline{R} = \widetilde{R}$$

*Proof of Lemma 5.* The distribution of $\widetilde{X}$ in the Problem 2 is equivalent to $X\Sigma^{1/2}$, where $X$ is as in Problem 1. Thus, the two settings are equivalent by the change-of-variable $\beta = \Sigma^{1/2}z$. Specifically,

$$\hat{z}_\lambda := \operatorname*{argmin}_{z} ||\widetilde{X}z - \vec{y}||_2^2 + \lambda ||z||_{\Sigma^{1/2}M\Sigma^{1/2}}^2 \tag{56}$$

$$= \operatorname*{argmin}_{z} ||X\Sigma^{1/2}z - \vec{y}||_2^2 + \lambda z^T \Sigma^{1/2}M\Sigma^{1/2}z \tag{57}$$

$$= \operatorname*{argmin}_{z} ||X\Sigma^{1/2}z - \vec{y}||_2^2 + \lambda z^T \Sigma^{1/2}M\Sigma^{1/2}z \tag{58}$$

$$= \Sigma^{-1/2} \operatorname*{argmin}_{\beta=\Sigma^{1/2}z} ||X\beta - \vec{y}||_2^2 + \lambda \beta^T M\beta \tag{59}$$

Further, the response $\langle z^*, \widetilde{x} \rangle = \langle \beta, x \rangle$, and the test risk transforms identically:

$$\widetilde{R} = \mathop{\mathbb{E}}_{\widetilde{X}, y} [||\hat{z} - z^*||_{\Sigma}^2] + \sigma^2 \tag{60}$$

$$= \mathop{\mathbb{E}}_{X, y} [||\hat{\beta} - \beta^*||_2^2] + \sigma^2 \tag{61}$$

$$= \overline{R} \tag{62}$$

$$\square$$

This implies that if the covariance $\Sigma$ is known, then ridge regression with a $\Sigma^{-1}$ regularizer is sample-monotonic.

**Theorem 5.** *For all $n \in \mathbb{N}, d \in \mathbb{N}, \sigma \in \mathbb{R}$, covariance $\Sigma \in \mathbb{R}^{d \times d}$, and ground-truths $\beta^* \in \mathbb{R}^d$, the following holds.*

*Suppose $n$ samples $(x, y)$ are drawn with covariates $x \sim \mathcal{N}(0, \Sigma)$ and response $y = \langle \beta^*, x \rangle + \mathcal{N}(0, \sigma^2)$. Let $X \in \mathbb{R}^{n \times d}$ be the matrix of covariates, and $\vec{y}$ the vector of responses. For $\lambda > 0$, consider the ridge regression estimator with $\Sigma^{-1}$-regularizer:*

$$\hat{\beta}_{n, \lambda} := \operatorname*{argmin}_{\beta} ||X\beta - \vec{y}||_2^2 + \lambda ||\beta||_{\Sigma^{-1}}^2 \tag{63}$$

*Let $\overline{R}(\hat{\beta}_{n, \lambda}) := \mathbb{E}_{\hat{\beta}} ||\hat{\beta} - \beta^*||_{\Sigma} + \sigma^2$ be the expected test risk of the above estimator. Let $\lambda_n^{\mathrm{opt}}$ be the optimal ridge parameter (that achieves the minimum expected risk) given $n$ samples:*

$$\lambda_n^{\mathrm{opt}} := \operatorname*{argmin}_{\lambda} \overline{R}(\hat{\beta}_{n, \lambda})) \tag{64}$$

*And let $\hat{\beta}_n^{\mathrm{opt}}$ be the estimator that corresponds to the $\lambda_n^{\mathrm{opt}}$. Then, the expected test risk of optimally-regularized linear regression is monotonic in samples:*

$$\overline{R}(\hat{\beta}_{n+1}^{\mathrm{opt}}) \leq \overline{R}(\hat{\beta}_n^{\mathrm{opt}})$$

*Proof.* This follows directly by applying the reduction in Lemma 5 for $M = I_d$ to reduce to the isotropic case, and then applying the monotonicity of isotropic regression from Theorem 1. $\square$

### A.4 Additional Plots

We apply random features to Fashion-MNIST Xiao et al. (2017), an image classification problem with 10 classes. Input images $x \in \mathbb{R}^d$ are normalized and flattened to $[-1, 1]^d$ for $d = 784$. Class labels are encoded as one-hot vectors $y \in \{\vec{e_1}, \dots \vec{e_{10}}\} \subset \mathbb{R}^{10}$.

## B Towards Monotonicity with General Covariates

Here we investigate whether monotonicity provably holds in more general models, inspired by the experimental results. As a first step, we consider Gaussian (but not isotropic) covariances and homeostatic noise. That is, we consider ridge regression in the setting of Section 2, but with $x \sim \mathcal{N}(0, \Sigma)$, and $y \sim \langle x, \beta^* \rangle + N(0, \sigma^2)$. In this section, we observe that ridge regression can be made sample-monotonic with a modified regularizer. We also conjecture that ridge regression is sample-monotonic without modifying the regularizer, and we outline a potential proof strategy along with numerical evidence.

### B.1 Adaptive Regularization

The results on isotropic regression in Section 2 imply that ridge regression can be made sample-monotonic even for non-isotropic covariates, if an appropriate regularizer is applied. Specifically, the appropriate regularizer depends on the covariance of the inputs. For $x \sim \mathcal{N}(0, \Sigma)$, the following estimator is sample-monotonic for optimally-tuned $\lambda$: $\hat{\beta}_{n, \lambda} := \operatorname{argmin}_{\beta} ||X\beta - \vec{y}||_2^2 + \lambda ||\beta||_{\Sigma^{-1}}^2$. This follows directly from Theorem 1 by applying a change-of-variable; full details of this equivalence are in Section A.3. Note that if the population covariance $\Sigma$ is not known, it can potentially be estimated from unlabeled data.

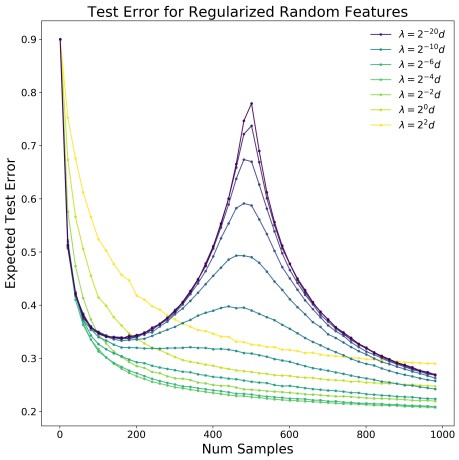

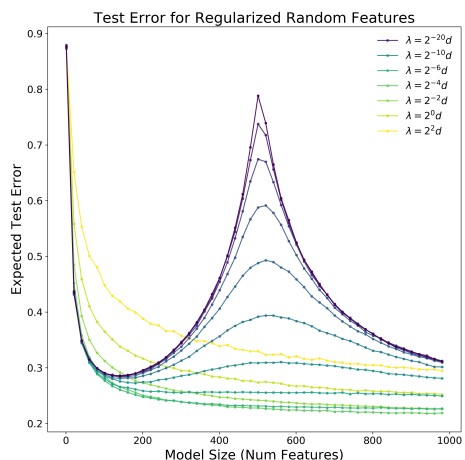

(a) Test Classification Error vs. Number of Training Samples.

(b) Test Classification Error vs. Model Size (Number of Random Features).

Figure 4: **Double-descent for Random ReLU Features.** Test classification error as a function of model size and sample size for Random ReLU Features on Fashion-MNIST. Left: with $D = 500$ features. Right: with $n = 500$ samples. See Figures 7, 8 for the corresponding test Mean Squared Error. See Appendix D of Nakkiran et al. (2020) for the performance of these unregularized models plotted across Num. Samples $\times$ Model Size simultaneously.

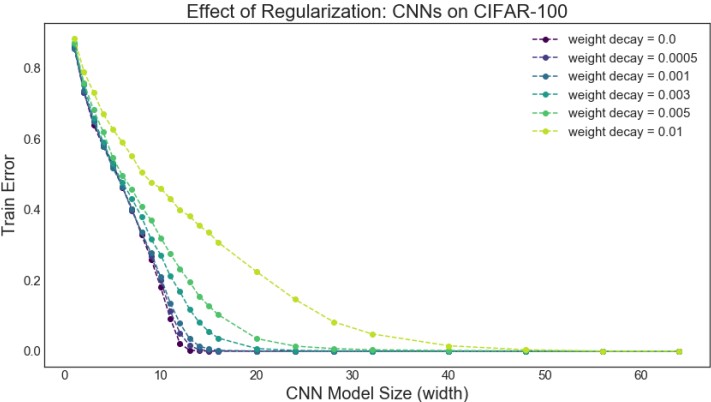

Figure 5: Train Error vs. Model Size for 5-layer CNNs on CIFAR-100, with $\ell_2$ regularization (weight decay).

### B.2 TOWARDS PROVING MONOTONICITY

We conjecture that optimally-regularized ridge regression is sample-monotonic for non-isotropic covariates, even without modifying the regularizer (as suggested by the experiment in Figure 2). We derive a sufficient condition for monotonicity, which we have numerically verified in a variety of instances. Specifically, we conjecture the following.

**Conjecture 1.** *For all $d \in \mathbb{N}$, and all PSD covariances $\Sigma \in \mathbb{R}^{d \times d}$, consider the distribution on $(x, y)$ where $x \sim \mathcal{N}(0, \Sigma)$, and $y \sim \langle x, \beta^* \rangle + \mathcal{N}(0, \sigma^2)$. Then, we conjecture that the expected test risk of the ridge regression estimator: $\hat{\beta}_{n,\lambda} := \operatorname{argmin}_\beta ||X\beta - \vec{y}||_2^2 + \lambda ||\beta||_2^2$ for optimally-tuned $\lambda \geq 0$, is monotone non-increasing in number of samples $n$. That is, for all $n \in \mathbb{N}$,*

$$\inf_{\lambda \geq 0} \overline{R}(\hat{\beta}_{n+1,\lambda}) \ \leq \ \inf_{\lambda \geq 0} \overline{R}(\hat{\beta}_{n,\lambda}) \tag{65}$$

*where we define $\hat{\beta}_{n,0} := \lim_{\lambda \to 0+} \hat{\beta}_{n,\lambda} = X^\dagger y$.*

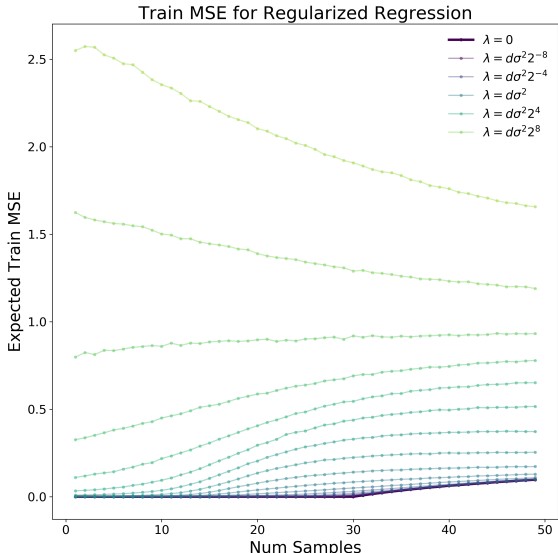

Figure 6: Train MSE vs. Num. Samples for Non-Isotropic Ridge Regression in $d = 30$ dimensions, in the setting of Figure 2. Plotting train MSE: $\frac{1}{n}||X\hat{\beta} - \vec{y}||_2^2$.

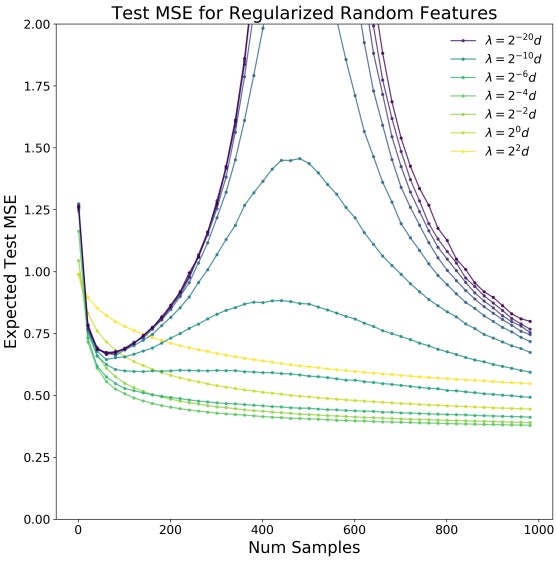

Figure 7: Test Mean Squared Error vs. Num Train Samples for Random ReLU Features on Fashion-MNIST, with $D = 500$ features.

In Appendix B.3 we present a technical conjecture in random matrix theory (Conjecture 2) which suffices to prove Conjecture 1. Proving this Conjecture 2 presents a number of technical challenges, but we have numerically verified it in a variety of cases. It can also be shown that Conjecture 2 is true when $Q = I$, corresponding to isotropic covariates. We prove the reduction between Conjecture 2 and 1 in Appendix B.3.

### B.3 MONOTONICITY CONJECTURE PROOFS

In order to establish Conjecture 1, it is sufficient to prove the following technical conjecture.

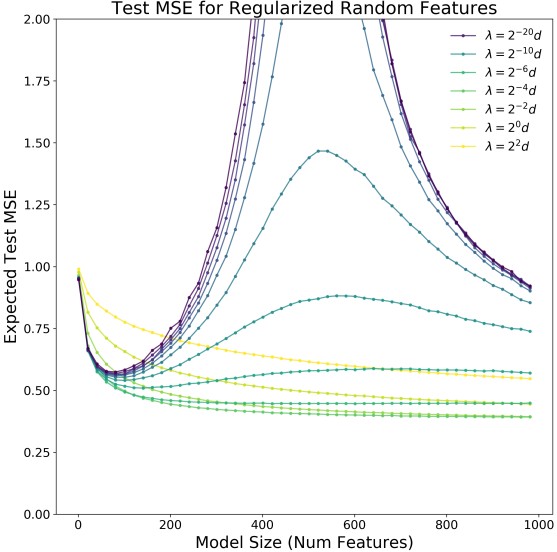

Figure 8: Test Mean Squared Error vs. Num Features for Random ReLU Features on Fashion-MNIST, with $n = 500$ samples.

**Conjecture 2.** *For all $n \in \mathbb{N}$, $d \geq n$, $\lambda > 0$, symmetric positive definite matrix $Q \in \mathbb{R}^{d \times d}$, the following holds.*

*Define*

$$G_\lambda^n := \lambda^2 \mathop{\mathbb{E}}_X [(X^T X + \lambda Q)^{-2}]$$

*where $X \in \mathbb{R}^{n \times d}$ is sampled with each entry i.i.d. $\mathcal{N}(0, 1)$. Similarly, define*

$$H_\lambda^n := \mathop{\mathbb{E}}_X [||(X^T X + \lambda Q)^{-1} X^T||_F^2]$$

*The expected test risk for $n$ samples can be expressed as:*

$$\overline{R}(\hat{\beta}_{n,\lambda}) = (\beta^*)^T G_\lambda^n \beta^* + \sigma^2 H_\lambda^n + \sigma^2 \tag{66}$$

*Then, we conjecture that the following two conditions hold.*

*1.*

$$G_\lambda^n \succeq G_\lambda^{n+1} \tag{67}$$

*2.*

$$(G_\lambda^n - G_\lambda^{n+1}) - (H_\lambda^n - H_\lambda^{n+1}) \frac{dG_\lambda^n/d\lambda}{dH_\lambda^n/d\lambda} \succeq 0 \tag{68}$$

**Lemma 6.** *Conjecture 2 implies Conjecture 1.*

*Proof.* By the reduction in Section A.3, showing monotonicity for non-isotropic regression with an isotropic regularizer is equivalent to showing monotonicity for isotropic regression with a non-isotropic regularizer. Thus, we consider the latter. Specifically, Conjecture 1 is equivalent to showing monotonicity for the estimator

$$\hat{\beta}_{n,\lambda} := \mathop{\text{argmin}}_\beta ||X\beta - \vec{y}||_2^2 + \lambda ||\beta||_{\Sigma^{-1}}^2 \tag{69}$$

$$= (X^T X + \lambda \Sigma^{-1})^{-1} X^T y \tag{70}$$

where $x \sim \mathcal{N}(0, I)$ is isotropic, and $y \sim \langle x, \beta^* \rangle + \mathcal{N}(0, \sigma^2)$.

Now, letting $Q := \Sigma^{-1}$, the expected test risk of this estimator for $n$ samples is:

$$
\begin{aligned}
\overline{R}(\hat{\beta}_{n,\lambda}) &= \mathbb{E}_{X,y}[||\hat{\beta}_{n,\lambda} - \beta^*||_2^2] + \sigma^2 \\
&= \mathbb{E}_{X,y}[||(X^T X + \lambda Q)^{-1} X^T y - \beta^*||_2^2] + \sigma^2 \\
&= \mathbb{E}_{X,\eta \sim \mathcal{N}(0,\sigma^2 I_n)}[||(X^T X + \lambda Q)^{-1} X^T (X\beta^* + \eta) - \beta^*||_2^2] + \sigma^2 \\
&= \mathbb{E}_X[||(X^T X + \lambda Q)^{-1} X^T X \beta^* - \beta^*)||_2^2] + \sigma^2 \mathbb{E}_X[||(X^T X + \lambda Q)^{-1} X^T||_F^2] + \sigma^2 \\
&= \mathbb{E}_X[||(X^T X + \lambda Q)^{-1}(X^T X + \lambda Q - \lambda Q)\beta^* - \beta^*)||_2^2] + \sigma^2 \mathbb{E}_X[||(X^T X + \lambda Q)^{-1} X^T||_F^2] + \sigma^2 \\
&= \lambda^2 \mathbb{E}_X[||(X^T X + \lambda Q)^{-1} Q \beta^*||_2^2] + \sigma^2 \mathbb{E}_X[||(X^T X + \lambda Q)^{-1} X^T||_F^2] + \sigma^2 \\
&= (\beta^*)^T G_\lambda^n \beta^* + \sigma^2 H_\lambda^n + \sigma^2
\end{aligned}
$$

Consider the infimum

$$
\inf_{\lambda \geq 0} \overline{R}(\hat{\beta}_{n,\lambda}) \tag{71}
$$

We consider several cases below.

**Case (1).** Suppose the infimum in Equation 71 is achieved in the limit $\lambda \to +\infty$. In this case, monotonicity trivially holds, since

$$
\lim_{\lambda \to \infty} \overline{R}(\hat{\beta}_{n,\lambda}) = \overline{R}(\vec{0}) = \lim_{\lambda \to \infty} \overline{R}(\hat{\beta}_{n+1,\lambda})
$$

**Case (2).** Suppose the infimum in Equation 71 is achieved by some $\lambda = \lambda_n^{\text{opt}}$ in the interior of the set $(0, \infty)$.

Because $\overline{R}(\hat{\beta}_{n,\lambda})$ is continuous and differentiable in $\lambda$ for all $\lambda \in (0, \infty)$, we have that $\lambda_n^{\text{opt}}$ must satisfy the following first-order optimality condition:

$$
\frac{d\overline{R}(\hat{\beta}_{n,\lambda})}{d\lambda}\bigg|_{\lambda=\lambda_n^{\text{opt}}} = 0 \tag{72}
$$

$$
\implies (\beta^*)^T \frac{dG_\lambda^n}{d\lambda} \beta^* + \sigma^2 \frac{dH_\lambda^n}{d\lambda}\bigg|_{\lambda=\lambda_n^{\text{opt}}} = 0 \tag{73}
$$

We will later use this condition to show monotonicity.

**Case (3).** Suppose the infimum in Equation 71 is achieved at $\lambda_n^{\text{opt}} = 0$. Recall, we define $\hat{\beta}_{n,0} := \lim_{\lambda \to 0+} \hat{\beta}_{n,\lambda}$. This means that,

$$
\frac{d\overline{R}(\hat{\beta}_{n,\lambda})}{d\lambda}\bigg|_{\lambda=0} = (\beta^*)^T \frac{dG_\lambda^n}{d\lambda} \beta^* + \sigma^2 \frac{dH_\lambda^n}{d\lambda}\bigg|_{\lambda=0} \geq 0 \tag{74}
$$

Note that since $\frac{dH_\lambda^n}{d\lambda} \leq 0$, both Equations (73) and (74) in Case (2) and Case (3) respectively imply that

$$
\sigma^2 \leq -\frac{(\beta^*)^T (\frac{dG_\lambda^n}{d\lambda})\beta^*}{dH_\lambda^n/d\lambda}\bigg|_{\lambda=\lambda_n^{\text{opt}}} \tag{75}
$$

Now, assuming Conjecture 2, we will show that the choice of $\lambda_n^{\text{opt}}$ in Cases (2) and (3) has non-increasing test risk for $(n+1)$ samples. That is,

$$
\overline{R}(\hat{\beta}_{n,\lambda_n^{\text{opt}}}) \geq \overline{R}(\hat{\beta}_{n+1,\lambda_n^{\text{opt}}})
$$

This implies the desired monotonicity, since $\overline{R}(\hat{\beta}_{n+1,\lambda_n^{\text{opt}}}) \geq \overline{R}(\hat{\beta}_{n+1,\lambda_{n+1}^{\text{opt}}})$.

We first consider the case when $H_\lambda^n - H_\lambda^{n+1}|_{\lambda=\lambda_n^{\text{opt}}} \geq 0$. In this case, because $G_\lambda^n - G_\lambda^{n+1} \succeq 0$ by assumption, we have

$$\overline{R}(\hat{\beta}_{n,\lambda_n^{\text{opt}}}) - \overline{R}(\hat{\beta}_{n+1,\lambda_n^{\text{opt}}}) = (\beta^*)^T(G_\lambda^n - G_\lambda^{n+1})\beta^* + \sigma^2(H_\lambda^n - H_\lambda^{n+1})\Big|_{\lambda=\lambda_n^{\text{opt}}} \tag{76}$$

$$\geq 0 \tag{77}$$

Otherwise, assume. $H_\lambda^n - H_\lambda^{n+1}|_{\lambda=\lambda_n^{\text{opt}}} \leq 0$. Then we have:

$$\overline{R}(\hat{\beta}_{n,\lambda_n^{\text{opt}}}) - \overline{R}(\hat{\beta}_{n+1,\lambda_n^{\text{opt}}}) = (\beta^*)^T(G_\lambda^n - G_\lambda^{n+1})\beta^* + \sigma^2(H_\lambda^n - H_\lambda^{n+1})\Big|_{\lambda=\lambda_n^{\text{opt}}} \tag{78}$$

$$\geq (\beta^*)^T(G_\lambda^n - G_\lambda^{n+1})\beta^* - (\beta^*)^T(\frac{dG_\lambda^n}{d\lambda})\beta^* \frac{(H_\lambda^n - H_\lambda^{n+1})}{dH_\lambda^n/d\lambda}\Big|_{\lambda=\lambda_n^{\text{opt}}}$$

$$\text{(by Equation (75), and } H_\lambda^n - H_\lambda^{n+1} \leq 0)$$

$$= (\beta^*)^T \underbrace{\left((G_\lambda^n - G_\lambda^{n+1}) - (H_\lambda^n - H_\lambda^{n+1})\frac{dG_\lambda^n/d\lambda}{dH_\lambda^n/d\lambda}\right)\Big|_{\lambda=\lambda_n^{\text{opt}}}}_{\succeq 0 \text{ by Conjecture 2}} \beta^* \tag{79}$$

$$\geq 0 \tag{80}$$

as desired.

$\square$

