# OpenReview forum: "Optimal Regularization can Mitigate Double Descent"
_ICLR.cc/2021/Conference — ICLR 2021 Poster_

### Official Review · AnonReviewer1 · 2020-10-28
**An interesting paper with neat results but potentially limited impact**

**Rating:** 7
**Confidence:** 3

**Review:**

The authors consider the problem of double descent in regularized linear regression estimators, and show that, with optimal regularization, such problems can be totally mitigated when the covariates are isotropic gaussian. Additionally, the authors show that there exist cases where ridge regression is provably non-monotonic in its average risk. These theoretical results are complemented with some empirical results on ridge regression and small scale neural networks.

The phenomenon of double descent has gathered substantial interest in the past couple of years, and this paper presents an interesting contribution towards cementing our understanding in the context of double descent in linear models. Although the obtained results are perhaps not overly surprising given the strong connection between the ridge estimator and linear regression with isotropic gaussian variables and homoskedastic gaussian noise, the finite-sample proofs characterize the phenomenon in a concise manner. The paper is clearly written, and exposes the results in a clear and accessible fashion. However, the impact of this paper is limited by the fact that the general phenomenon of regularization avoiding double descent is well-known (if not necessarily in such an explicit setup, at least from considerations in the Bayes-risk or minimax-risk framework, and penalized regression in general), and that interest in double descent mostly stems from the existence of this behavior in unregularized settings.

Other notes: there are some typos (especially in the appendix):
P.16 regularzier -> regularizer
P. 17 homeostatic -> homoskedastic

=====================

Update after author response: I thank the authors for their responses. I broadly agree with the points brought up by the other reviewers, and despite some weaknesses brought up by various reviewers, this paper is a good contribution to the community. I have brought my score from 6 to 7.

---

> ### Author Response · Authors · 2020-11-20
> **Response to R1**
>
> Thanks for your useful feedback, and for recognizing the value in our contributions towards understanding double descent.
>
> Regarding your concern on the "regularization avoiding double-descent" being well-known: We emphasize that this is subtle, since it is not true as universally as you may expect. For example, Section 4 shows a simple example of a linear regression setting where regularization does *not* suffice for monotonicity. Moreover, note that the "instance-specific" notion of monotonicity that we desire is significantly stronger than minimax-style monotonicity (which is worst-case over problem settings). In particular, there exist minimax-optimal estimators which are *not* monotonic in our instance-specific sense.

---

### Official Review · AnonReviewer3 · 2020-10-28
**Marginally above acceptance threshold**

**Rating:** 6
**Confidence:** 3

**Review:**

This paper studies the double-descent phenomenon from a theoretical perspective. It proves that for certain linear regression tasks, optimally-tuned L-2 regularized model is able to achieve monotonic test performance as the sample size or the model size increases. Empirical results on the effect of L-2 regularization on the double-descent phenomenon are provided for more general models and tasks.


Pros:

+ The motivation of studying the double-descent phenomenon with optimal regularization is well-explained in the introduction. Connections and comparisons with existing related works are discussed clearly.

+ The presented theoretical results on the linear regression model are non-asymptotic, which is new and different from existing works. The theoretical results are discussed with enough details such that I can easily follow the Lemmas and Theorems.


Cons:

- My main concern is the generality of the results. The paper mainly focuses on a simplified linear regression model, where the response variable is linearly generated using some ground-truth parameters \beta^*. One question is whether the results apply to the agnostic settings (the relationship between x and y is unknown)? Moreover, going beyond regression tasks, can we draw the same conclusion for classification tasks?

- The experiments need to be more extensive and better-explained, especially for the CIFAR-100 experiments. This is an image classification task, which is different from previously introduced regression setting. It is important to discuss this difference clearly at the beginning. In addition, the author uses a 5-layer CNN architecture, whereas the state-of-the-art CIFAR-100 results adopts a much larger architecture. Varying architecture size and even considering more image benchmarks would strengthen the experiments.


Minor Comments:

1. For Figure 3, it seems that for certain model size, the optimally-regularized test error is much lower than any other one. How do you compute the optimally regularized curve in Figure 3?

===== Post-Discussion Update =====

I thank the authors' efforts for responding my questions. Overall, I find the results presented in the paper interesting and worth publishing. It would be nice to extend the results to more general settings.

---

> ### Author Response · Authors · 2020-11-20
> **Response to R3**
>
> Thank you for your valuable feedback, and for appreciating the motivation of our work and the technical contribution of non-asymptotic results.
>
> We address your concerns and questions below:
> - Regarding generality, and "whether the results apply to the agnostic settings": We agree that it would be great to extend these results past the linear models considered in this work. One concern with even seemingly-minor extensions (such as the agnostic case) is that we do not fully understand even well-specified linear regression without the isotropic condition, and we have reason to believe that this is highly nontrivial. Specifically, proving monotonicity for non-isotropic covariates is still open (Conjecture 1 in the Appendix), and we have reason to believe that proving this requires highly nontrivial statements in random matrix theory (Conjecture 2). The difficulty in these settings arises because we desire **non-asymptotic** results, which would be much stronger than their asymptotic counterparts. Some versions of these questions are addressed in asymptotic settings in recent works of [Mei-Montanari https://arxiv.org/abs/1908.05355] and [d'Ascoli et al. https://arxiv.org/abs/2003.01054].
>
> - "can we draw the same conclusion for classification tasks?": We did not theoretically analyze classification settings, but we did test monotonicity for experimental classification settings in Section 5. It may be possible to analyze optimal regularization in **asymptotic** classification settings, building on recent works such as [https://arxiv.org/abs/1911.01544] and [https://arxiv.org/abs/2001.11572]. However, these techniques do not directly apply in the non-asymptotic settings considered here.
>
> - Thank you for the feedback on the experimental section; we will improve the clarity of the presentation.
>
> - "5-layer CNN": Note that while this 5-layer CNN is not a ResNet, it is in fact capable of reaching close to state-of-the-art accuracies on CIFAR-10/100, and is likely to have similar behavior as ResNets. In particular, this network was used in "Deep Double Descent" [https://arxiv.org/abs/1912.02292] where it was observed to have similar behavior as ResNet18s. This same network has also been used in works [https://arxiv.org/abs/2003.02237] and [https://arxiv.org/abs/2010.08508]. Thus, we believe that conclusions reached using this 5-layer CNN will carry over to larger and deeper networks as well.
>
> - "Figure 3": We apologize for the unclear figure labeling here. In Figure 3, the optimal regularization value (pink highlight) is always selected as one of the 6 values of regularization in the legend. The pink highlight overlaps the other lines in the figure, which unfortunately obscures them (we will fix this in the revision). For example, for model widths < 25, the optimal regularization is lambda=0.005, and then it reduces to lambda=0.003, etc.
>
> We hope we have addressed your questions; please let us know if there are any additional concerns.

---

### Official Review · AnonReviewer4 · 2020-10-29
**Fundamental research question, insightful theoretical and experimental results**

**Rating:** 7
**Confidence:** 4

**Review:**

This paper studies the double descent phenomena -- for increasing sample/model size -- in linear ridge regression. When the model is well-specified and the features are drawn from an isotropic Gaussian distribution, for the optimal (or larger than optimal) ridge regularization parameter, the authors show that there will be no double descent with respect to the sample size, i.e. the test error monotonically decreases as a function of the sample size. Furthermore, under a (randomly) projected feature space model, they showed that the test performance is also monotone with respect to projection size. The authors verify some of the the theoretical findings in practice, and show that the same insights carry over for more complicated models.

++:
The paper studies a fundamental question in theoretical machine learning.
The claims are supported both by clean theoretical results as well as empirical evaluations.

--:
The proof techniques seem to heavily depend on the specific choice of the loss function and the regularizer, that is, the mean squared loss and the ridge penalty. It is not clear if the techniques can generalize to other settings.

Question / Minor Comment:
The paper puts forward the idea that double descent is an artifact of underregularization. To be more precise, the main takeaway from the paper is that optimal \ell_2-regularization can mitigate double descent, provably in certain linear ridge regression problems; and in practice, in certain deep learning problems. A natural question is if authors have observed similar phenomena under different regularization techniques, including other norm-based penalties, or the more exotic ones from the deep learning literature.
I suggest that in the experiments section, you clearly state how you obtain the optimal regularization parameter in each of the subsections.
Why does the experiments section only cover the “sample monotonicity” part of the theoretical results (Theorems 1 and 2)? I think it helps if you verify Theorem 3 as well.
Can you comment on the requirement d <= p in Theorem 3? It’s true that d > p will not give a subspace of the p-dimensional ambient space, but one can perhaps extend the setting to d > p requiring that P^T P = I_p.

Overall, I enjoyed reading the paper and recommend it for acceptance in ICLR. The paper is well-written for the most part. I found the theoretical results insightful, and well-supported by experiments.



#####################################
I have considered the rebuttal as well as other comments in my final recommendation.

---

> ### Author Response · Authors · 2020-11-20
> **Response to R4**
>
> Thank you for your feedback, and for your helpful suggestions on the presentation. We will incorporate your suggestions in the revision.
>
> To answer your questions:
> - "Do [we] observe similar phenomena under different regularization techniques": We agree it is a good question to ask if other kinds of regularization / capacity-control can also eliminate double-descent in the same way. We expect that this will indeed be the case empirically (for many standard regularizers like dropout, etc), though we have not extensively tested this. Understanding the situation theoretically in these settings may also be much more difficult. However, we do have some evidence from prior works that other regularizers can also induce monotonicity. For example, if we consider early-stopping as a form of regularization, then Figure 1 of "Deep Double Descent" [https://arxiv.org/abs/1912.02292] shows that optimal early-stopping can empirically eliminate the double-descent in some settings (though this is not the focus of that work).
>
> - "Why does the experiments section only cover the “sample monotonicity” part of the theoretical results?": Note that Section 5.2 includes experiments on model-size monotonicity, to test nonlinear versions of Theorem 3 in practice. (This includes experiments on random-feature models and CNNs).
>
> - "the requirement d <= p in Theorem 3": You are correct that the theorem could be mathematically extended to d > p using similar techniques. However, this regime is quite different, in that it does not capture the "right" notion of increasing model size. Specifically, in the "d <= p" regime, increasing the "model size" by increasing 'd' makes the model more powerful, and achieves smaller test risk. However, once d > p, then all subsequent model sizes (d+1, d+2,...) are essentially the same model, and will have the same test risk as the d=p model. This is because for d > p, we can consider the model restricted to the p-dimensional image of P, and the d-dimensional model is equivalent to learning a p-dimensional model within this image. Intuitively, increasing 'd' once d > p is no longer giving the model more information about the covariates, but it's just changing the representation of the space.

---

### Official Review · AnonReviewer2 · 2020-11-01
**Nice paper addressing the role of regularization in mitigating double descent**

**Rating:** 7
**Confidence:** 3

**Review:**

he paper studies the surprising phenomenon of “double descent” in machine learning models which has recently come into light through many prior works. The phenomenon is used to describe the behavior of test performance of an estimator as the model parameters (complexity) or the number of samples are increased. It is observed that in many cases, the test error first decreases then increases attaining a peak and then starts to decrease again as the number of model parameters (or samples) are increased. This pervades many different models including neural networks, decision trees and linear regression.
Prior works have noted that this occurs primarily for unregularized or under-regularized models and that leads to the motivating question for this work: can optimal regularization remove double-descent? The paper studies this question from a theoretical and empirical perspective.
Theoretical: For the setting of linear regression the authors show that an optimal amount of l2 regularizer added to the objective completely removes the double descent phenomenon with respect to both number of samples and number of model parameters.
Empirical: For random feature classifiers and convolutional neural networks, the authors empirically show that an optimal amount of l2 regularization removes the appearance of double descent curves.

In terms of the theoretical analysis provided in this paper, a highlight is that the results are non-asymptotic and the Theorems identify precise values of the different parameters when the double descent phenomenon disappears.
The theoretical results have the caveat that they apply only when the input data is Gaussian. If the data is not Gaussian, l2 regularization might not be able to remove double descent as the authors demonstrate via the means of a counterexample. Moreover, if the covariates are not isotropic, then the authors conjecture but cant prove that optimal l2 regularization suffices.

In my view, the paper takes an important question and analyzes it well from a theoretical angle and also provides empirical evidence to back up its main message in more complex models. The proofs are non-trivial and I think the paper adds value in improving our understanding of the double descent phenomenon by providing a clear picture of the non-asymptotic regime.


Questions:
1. If I recall prior works correctly, many of them point out that double descent continues to hold even under different regularization schemes both theoretically and empirically. What changed here or what were the prior works missing? I am asking this because based on prior work your result sends a conflicting message that we can get rid of double descent and also not sacrifice performance at the same time.
2. Curious to hear what happens if we look at deeper neural networks? Could the optimal regularization amount be so high that it starts to hurt performance?
3. Equation (9) in the Appendix would benefit from more explanation. In particular, the calculation showing how we arrive at the singular values of the matrix (X^TX + lambdaI)^-1X^T.

Minor typos:
1. Abstract: “quantities such as the …”
2. Extra bracket in definition of lambda_n^opt on page 4


-------------

Thank you to the authors for their response. It has helped clear some questions I had in mind. I am keeping my rating.

---

> ### Author Response · Authors · 2020-11-20
> **Response to R2**
>
> Thank you for your valuable feedback, and for recognizing the theoretical and empirical contributions in our work.
>
> To answer your questions:
> 1. "What changed here or what were the prior works missing?": Many of the prior works observed double-descent in settings where the model size or samples was increased, but all other hyperparameters (eg regularization strength, and train time) were held fixed. In contrast, the message of our work is that if regularization is tuned optimally **as a function of** model size -- which is more aligned with the situation in practice -- then double descent can be mitigated. At a high level, the takeaway from prior works on double descent was that "if you are not careful, you could encounter pathological behaviors near the critical regime". While the message of our work is "if you **are** careful (i.e. you optimally regularize), then you can avoid these pathologies."
>
> 2. "Could the optimal regularization amount be so high that it starts to hurt performance?". Great question. On the one hand, in theory we always include \lambda=0 as a possibile regularizer, so "optimal regularization" is always at least as good as no regularization in terms of performance. However, you are right that as we look at networks with larger capacity, the optimal regularization value seems to tend to 0 in practice. This is evident in our Figure 3, and has also been observed in prior works (e.g. https://arxiv.org/abs/1805.10939).
>
> 3. "Equation (9) in the Appendix": Thanks for noticing this, we will make it more clear in the revision. Briefly, the $\gamma_i$s are eigenvalues of the matrix $X$ itself, and Equation (9) follows by using the SVD decomposition of $X = U \textrm{diag}(\gamma_i) V^T$ and the fact that U and V are orthonormal.

---

### Public Comment · ~Denny_Wu2 · 2020-11-15
**A few comments on prior works**

Thanks for the interesting work. A few comments:

+ The observation that ridge regularization suppresses the double descent peak is not new — it at least dates back to 1992 (and should therefore be cited):
Krogh, A. and Hertz, J.A., 1992. A simple weight decay can improve generalization. Advances in neural information processing systems.

+ The discussion of “triple descent” is rather misleading — such trend is not a unique feature of the non-asymptotic analysis; instead it can be easily obtained (in fact even more peaks) in the asymptotic setting of [Dobriban and Wager 2018] or [Hastie et al. 2019],  by manipulating the data covariance. This point needs to be clarified.

+ For isotropic data, the monotonicity of the optimally (ridge-) regularized prediction risk is a relatively well-known conclusion in the proportional limit (e.g. Proposition 6 in [Dobriban and Sheng 2019], but earlier reference could exist).
I therefore think that the current submission should be more explicit in describing (part of) its theoretical contribution as proving a non-asymptotic version of existing results on risk monotonicity.

+ Regarding general covariates, we recently showed that in the proportional limit, such monotonicity holds for the random-effect model studied in [Dobriban and Wager 2018]. But for more general orientation of the true parameters this result may be challenging to establish — note that in this case the optimal regularization strength $\lambda$ can be negative.
Wu, D. and Xu, J., 2020. On the optimal weighted $\ell_2$ regularization in overparameterized linear regression. Advances in neural information processing systems.

---

> ### Author Response · Authors · 2020-11-17
> **Response to Denny Wu**
>
> Thank you for your interest. We would like to clarify a few points. Our work makes very clear that our contribution is non-asymptotic; such a finite sample, non-asymptotic analysis is important to accurately characterize real phenomena (also see further related work below). Furthermore, we have cited all the references you mention, aside from [Krogh, A. and Hertz, J.A., 1992] and your concurrent work [Wu, D. and Xu, J., 2020], which we will add.
> In addition, we believe a number of your remarks are misleading for the following reasons:
>
> - We did include references as early as [Trunk 1979] as part of our list of works which have observed and studied this phenomena in some form; we are not claiming this is new. We will also add the reference to the work [Krogh, A. and Hertz, J.A., 1992] to this list.
>
> - We do not believe the discussion of "triple descent" is misleading, nor do we say that it is a consequence of non-asymptotic analysis. For works you referenced, we do not see a discussion (or experiments) exhibiting triple or multiple descent. If we are mistaken, can you please explicitly provide a reference in these papers? We would be happy to clarify this point because we do seek to credit this study in an accurate manner. The only prior work we are aware of that discusses this is [Liang et al. 2020], which we cite in context. Also, the work of [Chen et al] on multiple descent does not cite prior to [Liang et al. 2020] on this phenomena.
>
> - We clearly acknowledge the asymptotic related works in our paper, including [Dobriban and Sheng 2019] and [Dobriban and Wager 2018]. We are also very explicit about the non-asymptotic aspect of our contribution.
>
> Finally, note that several works which appeared subsequent to our work also discuss triple-descent, and acknowledge our contribution in isolating this phenomena, including:
>
> Multiple descent: Design your own generalization curve. Lin Chen, Yifei Min, Mikhail Belkin, Amin Karbasi. 2020
>
> Triple descent and the two kinds of overfitting: Where & why do they appear? Stéphane d'Ascoli, Levent Sagun, Giulio Biroli. 2020.

---

> > ### Public Comment · ~Denny_Wu2 · 2020-11-17
> > **Reply to authors**
> >
> > Thanks for the reply. To clarify a few things,
> >
> > + I agree that asymptotic risk characterizations on ridge regression (and double descent-related results) are properly cited.
> > What I intend to convey is that the *risk monotonicity* of optimally-tuned ridge regression is a known result in the proportional limit (under the same isotropic features that this submission assumes), and I do not see this fact being acknowledged in the related works (even though [Sheng and Dobriban 2019] is cited).
> > So while the statement that the current submission provides the first non-asymptotic characterization of such monotonicity is correct (to my knowledge), I think it should be highlighted that similar observations exist and have been rigorously proved in the asymptotic limit.
> >
> > + Similarly, I brought up [Krogh and Hertz 1992] not because it demonstrates sample-wise monotonicity (which I believe is the reason you're referring to the earlier works?), but rather that it indicates weight decay can *suppress* the double descent peak, which is very relevant to your message.
> >
> > + Regarding the "triple descent" risk curve, since this discussion is included in the "Universal vs Asymptotic" paragraph, my impression is you're suggesting that the presence of this trend is a benefit of the non-asymptotic setup, and would not be observed in the asymptotic analysis (which is *not true*); if this is not the case, sorry for the misinterpretation.
> > The reason that I mentioned [Dobriban and Wager 2018] is that under the general data covariance that they considered, it is very straightforward to construct multiple peaks (for instance see remark after Corollary 3 in [Wu and Xu 2020]), although the shape of the risk curve is not the concern of the original paper.

---

### Decision · Program_Chairs · 2021-01-07
**Final Decision**

**Decision:**

Accept (Poster)

**Comment:**

Quality: the paper takes an important question and analyzes it well from a theoretical angle; it also provides empirical evidence to back up its main message in more complex models. The proofs are non-trivial. The paper adds value in improving our understanding of the double descent phenomenon by providing a clear picture of the non-asymptotic regime.

Clarity: The motivation of studying the double-descent phenomenon with optimal regularization is well-explained in the introduction. Connections and comparisons with existing related works are discussed clearly. The paper is clearly written, and exposes the results in a clear and accessible fashion.

Originality: The presented theoretical results on the linear regression model are non-asymptotic, which is new and different from existing works.

Significance: The proof techniques seem to heavily depend on the specific choice of the loss function and the regularizer, that is, the mean squared loss and the ridge penalty. It is not clear if the techniques can generalize to other settings, which affects its significance.

Main Pros:
- the paper takes an important question and analyzes it well from a theoretical angle. The proofs are non-trivial; the paper adds value in improving our understanding of the double descent phenomenon by providing a clear picture of the non-asymptotic regime.

Main Cons:
- Generality of the results. The paper mainly focuses on a simplified linear regression model, where the response variable is linearly generated using some ground-truth parameters \beta^*.
- The experiments need to be more extensive and better-explained, especially for the CIFAR-100 experiments. It is important to discuss this difference clearly at the beginning.